# A Model for Combinatorial Dictionary Learning and Inference

**Avrim Blum**                                                          AVRIM@TTIC.EDU
*Toyota Technological Institute at Chicago, 6045 S. Kenwood Ave., Chicago, IL, 60637*

**Kavya Ravichandran**                                                 KAVYA@TTIC.EDU
*Toyota Technological Institute at Chicago, 6045 S. Kenwood Ave., Chicago, IL, 60637*

**Editors:** Gautam Kamath and Po-Ling Loh

## Abstract

We are often interested in decomposing complex, structured data into simple components that explain the data. The linear version of this problem is well-studied as dictionary learning and factor analysis. In this work, we propose a combinatorial model in which to study this question, motivated by the way objects occlude each other in a scene to form an image. First, we identify a property we call "well-structuredness" of a set of low-dimensional components which ensures that no two components in the set are *too* similar. We show how well-structuredness is sufficient for learning the set of latent components comprising a set of sample instances. We then consider the problem: given a set of components and an instance generated from some unknown subset of them, identify which parts of the instance arise from which components. We consider two variants: (1) determine the minimal number of components required to explain the instance; (2) determine the *correct* explanation for as many locations as possible. For the latter goal, we also devise a version that is robust to adversarial corruptions, with just a slightly stronger assumption on the components. Finally, we show that the learning problem is computationally infeasible in the absence of any assumptions.

**Keywords:** dictionary learning, learning structure from data, semi-random model

## 1. Introduction

The explosion of machine learning and statistical techniques for high-dimensional data has been met with the desire to find small but close-to comprehensive explanations for vast data. Indeed, we are often interested in developing good representations for complex data as sparse combinations of latent factors. Well-studied versions of this problem include matrix factorization, dictionary learning, and topic modeling. Most of these assume that the sparse combination of factors occurs linearly. However, in practice, many settings involve non-linear combinations of features. For example, an image can be described by referring to the objects in it and their locations. However, the objects are not combined *linearly* to produce the images. Rather, objects may be placed atop one other, causing occlusions. Similarly, in a musical medley (or a collage of images), we cut up pieces of the original objects (songs or images, respectively) and paste them together. These settings motivate studying dictionary learning in combinatorial and non-linear settings.

In this work, we study one such setting, inspired by one-dimensional images. So-called "images" are generated by first choosing components we call "objects" (analogous to the *atoms* of the dictionary in traditional dictionary learning) from a set, placing the chosen objects onto a canvas at chosen locations, and allowing objects placed more recently ("in front") to occlude those placed earlier ("in the back"). We refer to the canvas at the end of this process as the "image." Our goal is to give algorithms and guarantees for discovering the underlying objects from a collection of sample images. Then, we also wish to explain the composition of an image based on the learned set of objects, namely given an image, we would like to recover which objects are in it and where.

While we refer to simple components as "objects" and composite instances as "images," we want to emphasize that our goal is *not* to precisely model real-world image generation but rather to provide a combinatorial setup in which to theoretically study dictionary learning; images simply prove a familiar case where such an abstraction applies.

Additionally, we investigate adversarial robustness: knowing that the data were generated from fixed underlying factors through some process might help us better detect and correct adversarial perturbations. Our model provides a controlled setting in which to investigate adversarial robustness in this sense; we provide some results toward this.

Our main contributions are:

- we define a model for generating complex ("image") data from a small set of factors ("objects") combined in a non-linear way.
- we identify properties that suffice for learning in this model.
- we give algorithms to efficiently learn underlying factors (i.e., recover the dictionary) in various settings.
- we give algorithms to efficiently infer which objects out of a learned or otherwise specified set comprise a new image (we call this task "segmentation"). As part of this, we also address noise-tolerance, giving a few results about adversarial robustness in this model.

The rest of the paper is organized as follows. First, we summarize related work from the several areas we draw on. Next, in Section 2, we set up notation and precisely define the model which we study. Subsequently, we study how many samples are required to decompose "images" into "objects" in Section 3. Finally, in Section 4 we consider the segmentation problem under both the objective of determining a minimal explanation (Section 4.1) and the objective of getting a correct explanation on most of the image (Sections 4.2, 4.3). The lattermost section considers an adversarially noisy case.

**Related Work**   We briefly describe work related to ours in several different fields. For a more thorough discussion, please see Appendix A.

The issue of finding a small number of explanatory features for high-dimensional data is well-studied. Some particular framings of this problem include matrix factorization, for which principal components analysis (PCA) is often used (Karl Pearson, 1901; Hubert et al., 2000), dictionary learning, framed slightly more abstractly (Kreutz-Delgado et al., 2003), and factor analysis (Harman, 1960). While much of the work traditionally done in this area focuses on the linear case, there have been several works that address non-linear settings (Balcan et al., 2015, 2020; Mairal et al., 2009; Papyan et al., 2017). Several works have explored using the aforementioned methods to promote adversarial robustness (Bhagoji et al., 2018; Gupte et al., 2022). While much of existing theory in adversarial robustness is in classification, practical interest is often in segmentation.

Another line of work related to ours is that of factorial learning (e.g. Ghahramani, 1994) and layered models (e.g. Wang and Adelson, 1994). These decompose realistic images via inference over graphical models in contrast to our study of a stylized model where we prove guarantees.

Our work also draws heavily on problems studied in and techniques from computational biology. A well-studied problem in gene sequencing is reconstructing a string based on having seen recurring pieces of it (Motahari et al., 2013). We adapt such a shotgun sequencing algorithm for our use, as it will be applicable throughout Section 3.

## 2. Notation, Preliminaries, and Definitions

### 2.1. Notation

We study images that are produced on one-dimensional canvases of size $d$, which should be thought of as large. Formally, a *canvas* is a vector of pixels, where a pixel is an element of $[c]$, where $c$ is the number of colors. In the universe, there are $m$ objects, of which $k$ appear in any given image. We think of $k << m$. However, just because there are few objects in any image doesn't mean we get to easily see all of them. The size of object $i$, $s_i$, might be up to a constant fraction of the size of the canvas, so placing multiple objects into the canvas risks occlusion of one object by potentially several others. In particular, objects that tend to appear in the background would rarely be visible in full. Thus, the challenge of the learning part will be finding a set of pieces sufficient to reconstruct each object and then combining them correctly. We name the bounds on $s_i$ as $s_i \in [s_{\min}, s]$. Object $i$, is in $\{0, 1, \ldots, c-1\}^{s_i}$. In general, we suppose the background is a color distinct from the set of colors used for objects. When we refer to subsections of a string, we use the notation $\sigma[i:j]$ to denote the indices from $i$ to $j$, inclusive, of string $\sigma$.

**Assumptions**  In several sections, we specify lower bounds on the object size required in terms of the number of pixels required to uniquely identify an object (formally defined in the following section). The canvas size $d$ is much bigger than the size of signatures of objects (defined more formally later but essentially capture the shortest unique strings in the image) $w$: namely, $d > 8w$. Further, for learning, it is sufficient if an object is at most half the size of the canvas, i.e., $s < d/2$. For ease of notation, we let $d' = d + s - 2$. Also, in Section 3.2, $m \cdot d'$ must be bigger than a fixed constant, but in Section 3.3, we need a stronger condition, with $d' > O(m \, k \, 2^k)$.

### 2.2. Structural Properties and Object Generation

Next, we specify our setting further by identifying a property of objects that is useful for the learning task. We then show that random, and even semi-random, objects satisfy this property with high probability. This property (Defn. 1) helps guarantee that no two objects are "too similar," similar to incoherence assumptions in statistics, and so objects can be identified from signatures, pieces large enough to be uniquely associated with them. The stronger version (Defn. 2) is useful when adversarial corruptions are present.

**Definition 1 ($w$-well-structured)**  *We call a set of $m$ objects over the colors $\{0, 1, \ldots, c-1\}$ $w$-well-structured if: (1) Each object $o_i$ has length at least $w$; (2) No two objects have identical substrings of length $w$; and (3) No object has two identical substrings of length $w$.*

**Definition 2 ($\epsilon$-strongly, $w$-well-structured)**  *We call a set of $m$ objects $\epsilon$-strongly $w$-well-structured if: (1) Each object $o_i$ has length at least $w$; (2) No two objects have substrings of length $w$ that match in $1 - \epsilon$ fraction of their pixels; (3) No object has two substrings of length $w$ that match in $1 - \epsilon$ fraction of their pixels.*

When it is clear from context what $w$ is, we may elide it. These properties are natural, as objects generated randomly or semi-randomly satisfy them. We give the statement for random objects; see Appendix B.2 for the lemma for semi-random objects and proofs for both.

**Lemma 3** *[Random Objects are $w$-Well-Structured whp] A set of $m$ objects, each sampled uniformly at random from $\{0, 1, \ldots, c - 1\}^{s_i}$, and $s := \max_i s_i$ is $w$-well-structured with probability at least $1 - 3m^2 s^2 / c^w$. In particular, $w = O(\log ms)$ is sufficient so that the $m$ objects are $w$-well-structured with probability $1 - o(1)$.*

When the background is a single color, we define a $b$-padding of an object as an extension of the object created by adding $b$ pixels of background color on both left and right sides.

## 2.3. Image Generation

Next, we address how to generate images of size $d$ from these objects. In particular, we follow the following general high-level steps. In this section, we describe different models corresponding to different ways to make the choices below.

1. Set background to a color distinct from the colors in the objects (Defintion 4)[1].
2. Select $k$ objects in the uniform model (Definition 6).
3. Select $k$ indices randomly, representing locations in image for placing objects either according to the closed room (Definition 7) or open room (Definition 8) model.
4. Select depth index as determined by the depth model (Definitions 9 and 10).
5. Generate image by scanning left to right and, for each location, set the color of this position in the image to the color of the corresponding position in the topmost object. Equivalently, compute the `view` operator of the scene $\mathcal{S}$ given by the above, both defined at the end of this section.

In the first step above, we must choose how to set the background of an image.

**Definition 4 (Distinct Background)** *In the distinct background model, the background of the image is a value not in the alphabet $[c]$ of which the objects are composed.*

**Definition 5 ($w$-Well-Structured Background)** *In the well-structured background model, no substring of length $w$ of the background matches a different substring of length $w$ in the background or in any of the $m$ objects.*

Next, we define a uniform model for how objects are chosen to be in the image.

**Definition 6 (Uniform Model)** *In the* uniform model*, for each image, the set of objects in it is chosen uniformly over the $\binom{m}{k}$ sets of $k$-object subsets of the $m$ total objects.*

In the third step, we choose locations to place the objects on the canvas. Either all objects in the frame must be within the frame, or objects in the frame are allowed to spill off the edges. Formally:

**Definition 7 (Closed Room Model)** *In the* closed room model*, all objects begin and end within the canvas of size $d$. Formally, the left endpoint of an object $o_i$ is uniformly distributed over 0 to $d - s_i$.*

---

1. We could also consider the background to be $w$-well-structured along with the set of objects (Defintion 5); where results can be extended to this case, we note the differences in argument when they arise.

We call it this because it is reminiscent of items being placed within a room that is fully photographed, so objects are limited to not bleeding out of the scope of the camera. On the other hand, the canvas could be "open" at the boundaries:

**Definition 8 (Open Room Model)** *In the* open room model*, objects can begin and end off the canvas of size $d$ and we only consider what is visible in those $d$ locations. Formally, the left endpoint of an object $o_i$ is uniformly distributed over $-s_i + 1$ to $d$.*

Learning is similar in both models, so there we focus on the open room model, but in the inference problem, the kind of room matters. Finally, we describe the ways in which we choose the ordering of objects within the image. Locations are always chosen uniformly at random horizontally, but the depth at which an object is placed could involve ordering rules.

**Definition 9 (Fully Random Model: Random Horizontal; Random Depth)** *Objects are placed in random depth order on the canvas, at uniformly random positions left-to-right.*

**Definition 10 (Partially Random Model: Random Horizontal; Arbitrary Depth)** *The depth of objects placed on the canvas is given by arbitrary ordering constraints, but the horizontal placement of the objects is uniformly random.*

This more closely reflects what we might see in real life: certain objects may more consistently appear in the background – such as mountains or tall buildings.

The motivation behind our generative model is to study a form of *compression* or *dimensionality reduction* via sparse dictionary learning. Indeed, the image generation framework described here imposes rich structure on the space of images: if the images were completely random, there would be $c^d$ possible images, but here there are only $\binom{m}{k} \cdot k! \cdot d^k$ possibilities for a fixed set of objects.

**Defining the "scene" and "view"** To formally represent this procedure, we construct the matrix $\mathcal{S} \in \{[c] \cup \{b, \perp\}\}^{(k+1) \times d}$, which will represent the scene comprised of the chosen objects. We will then define a `view` operator that acts on a scene $\mathcal{S}$ to produce the image $\mathcal{I}$ that we study.

The matrix $\mathcal{S}$ is initialized with $\perp$ everywhere and constructed as follows. Row $k + 1$ is filled with the background chosen in step 1 above. Then, the objects are sorted based on the depths in step 4 above. The object with highest depth is placed in row $k$, starting at its respective location as chosen in step 3. This process is continued with the object with the lowest depth placed in the first row of the matrix. This gives us the scene matrix $\mathcal{S}$. Note that in the closed room model, each row of the matrix $\mathcal{S}$ either has $\perp$ at the start and end or the start or end of the object corresponding to that row, whereas in the open room model, the first elements of a row of $\mathcal{S}$ might be from the middle of an object, rather than $\perp$ or the leftmost pixels of an object and likewise for the last elements of the row.

**Definition 11** *For a sequence $S = \{(o_i, l_i)\}_{i=1}^{k}$ where $o_i$ is the object, $l_i$ is the location (left-right), and the index in the sequence defines the depth, the operator `scene`$(S)$ produces a matrix by padding $o_i$ with $\perp$ such that the string is $d$ in length and $o_i$ begins at location $l_i$ and places that into the row $i$ of the matrix.*

With the matrix defined, we can now define the `view` operator. We scan the $\mathcal{S}$ by column. At each column, we scan from top to bottom, picking out the first element that is not $\perp$ and placing that in the corresponding column of $\mathcal{I}$.

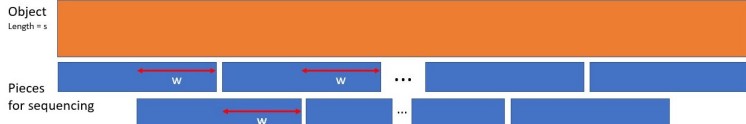

Figure 1: **Object recovery from pieces:** Due to objects obscuring each other, we cannot hope to see all $s$ pixels of the orange object at once, but so long as we see the blue pieces shown in this figure, i.e., ones with at least $w$ pixels overlap, we can reconstruct the object.

For instance, consider objects $10111$ (at depth 1, at position 1) and $0001$ (at depth 2, at position 4) and background $b, b, b, b, b, b, b, b$ :

$$\mathcal{S} = \begin{pmatrix} 1 & 0 & 1 & 1 & 1 & \perp & \perp & \perp \\ \perp & \perp & \perp & 0 & 0 & 0 & 1 & \perp \\ b & b & b & b & b & b & b & b \end{pmatrix}$$

Now, we take $\texttt{view}(\mathcal{S}) = \begin{pmatrix} 1 & 0 & 1 & 1 & 1 & 0 & 1 & b \end{pmatrix}$ .

### 2.4. Formal Statement of the Problem

**Learning:** Given a set of $S$ images, generated independently according to the process described in Section 2.3, reconstruct the set of $m$ objects that generated them.

**Inference:** Given a set of $m$ objects and a new image known to be generated according to the process described in Section 2.3, recover an explanation for the image, i.e., identify which pixel of which object produced each pixel of the image, subject to constraints of the generative process.

### 2.5. Sequencing

At a high level, our approach to learning objects from images will entail picking out pieces of images we can be reasonably certain come from a single object (we will call each of these a "segment") and then stitching them together to recover that object. Reconstructing a string on the basis of having seen pieces of it is a well-studied problem, particularly in gene sequencing (Motahari et al. (2013)). We adapt such a shotgun sequencing algorithm for our use and present it here, as it will be applicable throughout Section 3.

Assume our set of objects is $w$-well-structured. Suppose we are given a collection of segments, each of length at least $2w$ such that each one is a substring of one of those objects and moreover, for every object, every part of it has been seen by some segment in this set. We require adjacent pieces from the image to have $w$ pixels of overlap to uniquely identify which pieces to pair them with. Thus, if the length of an object is $s$ pixels, we require the specific $s/w$ segments (see Figure 1).

Then we can use a greedy algorithm to stitch together pieces of length $2w$ arising from objects that we find in the images to recover the full object. We search through the pieces we have collected to find the two with the largest overlap. We join them and continue this repeatedly.

The shotgun sequencing literature has shown that coverage is necessary and sufficient for recovery, i.e., seeing all pieces of the object with sufficient overlap between pieces suffices for reconstructing the object from those pieces (Thm. 1 in Motahari et al. (2013)). The greedy algorithm described above achieves this. Our proof of correctness relies on the sufficiency of coverage.

---

**Algorithm 1** Sequence

---
1: **Input:** set of segments of objects, signature length $w$
2: `objects_stored` $\leftarrow \{\}$
3: **while** segments remain **do**
4:     find two segments with longest overlap $\geq w$ pixels
5:     merge them at the overlap and add to `objects_stored`
6: **end while**
7: **return** `objects_stored`

---

**Proof of Correctness**   If the underlying objects are $w$-well-structured, and we have all required pieces of size $2w$ to cover each object completely, then the sequencing algorithm will reconstruct the objects exactly. This is because if $w$ pixels are sufficient as a signature for the object, then any set of $\geq w$ pixels that match between two strings must come from the same place in a single object. Further, once exactly $m$ pieces remain, we know that by well-structuredness, no two of them can share more than $w$ pixels and so no more merging will occur.

**Sample Complexity**   By the argument above, it is clear that for well-structured objects, as long as each object is covered, then this sequencing algorithm will recover the objects accurately. Thus, if $S$ samples suffice to cover the objects (and we will derive $S$ for the various settings in the respective sections), then $S$ samples suffice for learning the objects.

## 3. Learning

In this section, we address learning. We start with a simplified setting, where the start and end of objects are clearly marked, and we give an algorithm to learn the objects from a set of images and analyze how big this set must be. Next, we show that we can get guarantees even without endpoint markers when there are at most two objects in each image. Finally, having established much of the necessary analysis, we extend this to a setting where there are up to $k$ objects in each image, where $k$ is constant or grows sufficiently slowly with $d$ (size of image). Here, we assume the background is distinct. Further, we require that all objects in the set are of size at most $s$ and at least $O(w)$. We show in Appendix E that in the absence of any assumptions, the learning problem is NP-hard.

### 3.1. Learning in the presence of endpoint markers

Let us start by considering the case where every object comes with a clear and known left endpoint and a clear, distinct, and known right endpoint marker. For example, if objects are fully composed of red and blue, let each one have a yellow pixel appended to the left end and an orange pixel appended to the right end.

**Fully-Random Model**   As a warmup, we consider the Fully Random Model setting (Defn. 9) where objects with endpoint markers are placed in random order front-to-back. In this case, we

can be confident that at some point, each object will appear in the front. Since the left endpoint and right endpoint are marked differently, seeing a left endpoint marker, followed by many pixels, followed by a right endpoint marker implies we are seeing a full object. We simply wait to see all objects whole. The probability of a fixed object appearing in front is $\frac{k}{m} \cdot \frac{1}{k} = 1/m$, giving us that in expectation, we require $m$ samples to see it in the front. Using the coupon collector argument, if we see $\Theta(m \log m)$ samples, then with high probability, we will see each object in the front at least once.

We see that solving the problem in the case where each object has endpoint markers and a non-trivial chance of appearing at the front is actually quite simple. In the more general Partially Random Model, when objects may not appear in the front at all, we may only ever see certain objects with some amount of occlusion. Thus, one major part of the challenge arises from not seeing whole objects at once, and the other will arise once we no longer have endpoint markers.

**Partially-Random Model**  Next, we consider the Partially Random model (Defn. 10). Given images, recovery of a list of objects essentially requires finding chunks that belong to a single object. If a chunk is large enough, we can both uniquely identify it with an object (provided objects are well-structured) and find overlap with a different chunk that belongs to the same object. Algorithm 2 orders and joins such chunks. In the following lemma (proven in Appendix C.1), we compute the probability of seeing such a piece of interest.

**Lemma 12**  *In the open room, uniform object selection, partially-random model, the probability of seeing a given $L$-pixel segment of a given object (or even of an $(L-1)$-padding of that object) in a random image composed of $k$ objects is at least $\left(1 - \frac{s+L-1}{d'}\right)^{k-1} \cdot \frac{d+1-L}{d'} \cdot \frac{k}{m}$.*

This lemma is central to analyzing how many samples we need for the learning problem. In particular, we will use this lemma in the case where $L = 2w$, where $w$ is the well-structuredness parameter of the objects. In the following theorem, we analyze the number of samples required to succeed in the reconstruction effort with high probability.

---

**Algorithm 2** Recover Objects - With Endpoint Markers

---

1: **Input:** $S$ samples, value $L$
2: `chunks` $\leftarrow$ split images at markers (keeping track of what came with what marker)
3: `objects` $\leftarrow$ items from `chunks` that start with a start marker and end with an end marker of length $\geq L$
4: `object_pieces` $\leftarrow$ rest of the `chunks`
5: `objects` $\leftarrow$ `objects` $\cup$ SEQUENCE(`object_pieces`) {SEQUENCE given in Algorithm 1}

6: **return** `objects`

---

**Theorem 13**  *In the open room, uniform, partially-random model with endpoint markers and $w$-well-structured objects, given $S = \Theta(\ln(2\,m\,s/L)/a)$ samples, for $a = \left(1 - \frac{s+L-1}{d'}\right)^{k-1} \cdot \frac{d+1-L}{d'} \cdot \frac{k}{m}$ and $L = 2w$, Algorithm 2 reconstructs all objects accurately with probability $9/10$.*

**Proof Outline**  We show this theorem by first computing the probability of seeing a fixed $L$-pixel string from an object in a fixed image (Lemma 12). Then, we compute the probability of seeing

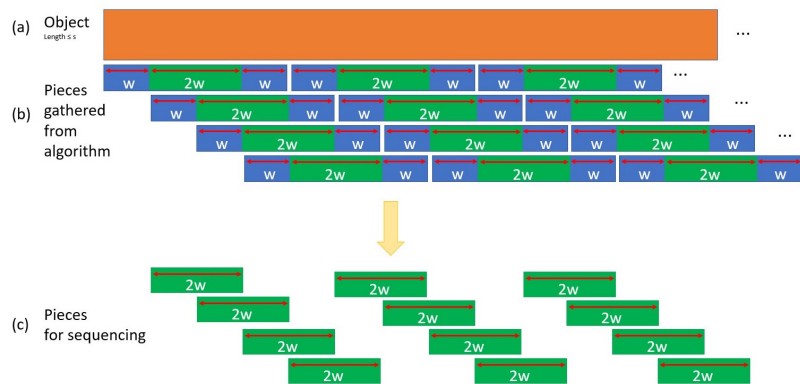

Figure 2: **Process used to learn objects from images in the absence of endpoint markers:** This algorithm allows us to collect the pieces of length $4w$ that cover the orange object in a redundant way. However, due to the risk of problematic overlaps we must discard a length-$w$ piece (blue) on either end. Even after this discard, the pieces (green) used for sequencing must cover the object.

it in any image and apply a union bound. See Appendix C.2 for details. From Figure 1, we can see that in this case, $L = 2w$ suffices for both having sufficient overlap between pieces to run the sequencing algorithm and each length-$L$ segment being uniquely identified to an object.

Note that the quantity $a$ drops exponentially with $k$, via the term $(1 - (s + L - 1)/d')^{k-1}$. Therefore, if $s + L$ is a constant fraction of $d'$, then the number of samples we need to observe according to Theorem 13 will be exponential in $k$. Indeed, this exponential dependence is unavoidable in the partially-random setting, because this is the sample size needed to see any given piece of an object that only appears in the back of an image. This result would also extend to the case where objects are $\epsilon, w$-strongly-well-structured, and at most $\epsilon/2$ fraction of the pixels are corrupted by an adversary before an object is placed. Details are in Appendix C.2.1.

### 3.2. No endpoint markers: Two Objects per Image

In general, we might not have endpoint markers. To analyze the sample complexity in that case, we first assume that there are at most two objects in an image. At a high level, our argument will be that for an appropriate value of $L$, length-$L$ strings that arise from a single object will occur much more frequently than length-$L$ strings that arise from pairs of overlapping objects when that overlap occurs in the middle half of the string. Therefore, we can safely gather the frequently-occurring length-$L$ strings and use the middle $L/2$ portions of them in sequencing Algorithm 1 (Figure 2).

To aid in analysis of Algorithm 3, we define the notion of a *problematic overlap*:

**Definition 14** *(problematic overlap, 2 object case) A window of length $L$ contains a* problematic overlap *if it contains two objects and no background, and they overlap inside the middle $L/2$ pixels. We call an $L$-pixel string a* problematic overlap string *if it can only arise via problematic overlaps.*

---

**Algorithm 3** Learn Objects – No Endpoint Markers

---

1: **Input:** $S$ samples, fraction $\tau$, value $L$
2: `object_pieces` ← middle $L/2$ locations of every length-$L$ string that appears at least $\tau \cdot S$ times and has no background
3: `objects` ← SEQUENCE(`object_pieces`) {SEQUENCE found in Algorithm 1}
4: **for** each object **do**
5:    `sleftend` ← leftmost $w$ pixels
6:    `leftend` ← shortest string $x$ in any image that has background to its left and `sleftend` to its right
7:    `srightend` ← leftmost $w$ pixels
8:    `rightend` ← shortest string $x$ in any image that has background to its right and `srightend` to its left
9:    append `leftend` to left of object and `rightend` to right of object
10: **end for**
11: **return** `objects`

---

**Theorem 15** *In the open room, uniform, partially-random model with $w$-well-structured objects, each of length at most $s$ and at least $4w$, and 2 objects per image, Algorithm 3 with $L = 4w$ recovers all $m$ objects with $S = \Omega\left(m \ln(ms)\right)$ samples and $\tau = \frac{1}{16m}$ with probability $\frac{9}{10}$.*

**Proof Outline**    To prove this, we must show three things.

1. **Seeing a particular big chunk from one object in one image is likely.** To argue about this, we simply apply Lemma 12 to compute the probability of seeing a fixed string in a fixed object, giving rise to Claim 25.

2. **Seeing a particular big chunk consisting of two objects (i.e., problematic overlap string) in a single image is unlikely.** In order to reason about this, we define the notion of a *problematic overlap* (Definition 14). Further, we argue that any string that contains a problematic overlap can only be generated in a small number of ways (Lemma 26). Finally, we use this to bound the probability that a particular problematic overlap string appears in a particular image, which we call $p_{\mathrm{bad}}$ (Claim 27).

3. **The separation between the aforementioned probabilities allows us to compute the sample complexity.** In Claim 28, we define a quantity $p_{\mathrm{mid}}$, that upper bounds the probability of seeing a problematic overlap string by a factor of 2 and lower bounds the probability of seeing a good string, again by a factor of two. The Chernoff bound allows us to determine the number of samples we require in order to see a sufficiently separation between the number of copies of good strings (Lemma 29). Thus, we compute the total number of good strings (arising from one object, Claim 30) and bad strings (arising from two-object problematic overlaps, Claim 31) and apply that lemma to get the theorem statement.

We need $L$ to be large relative to $w$, since we must ignore the ends of the length-$L$ string to avoid including problematic overlap strings in the sequencing. Figure 2 justifies the use of $L = 4w$. Finally, we also show that the set of samples contains all the information we need to recover the ends and that the procedure in lines 4-10 of the algorithm recovers them. For the full proof, see Appendix C.3.

### 3.3. Extension of argument above for several objects in image

We now consider the more general case where we can have up to $k$ objects in an image for some given value $k$ greater than 2. At a high level, our argument is as before. However, the major difference is that problematic-overlap strings (ones that were generated from combining two objects), can now be generated by up to $k$ objects. This increases our sample complexity. We once again assume objects are at most $s$ in size and $w$-well-structured, but now they must have length at least $6\,w\,k$.

**Theorem 16** *In the open room, uniform, partially-random model with $w$-well-structured objects, each of length at most $s$ and at least $6wk$, and $k$ objects per image, Algorithm 3 with $L = 8\,w\,k$ and $\tau = \frac{k}{O(m\,2^k)}$ recovers all $m$ objects with $S = \Omega\left(2^k\,m\ln(m\,s)\right)$ samples with probability 9/10.*

**Proof Outline** The first and third steps essentially carry over from the two object case, but analyzing the influence of problematic overlap strings in this case is more involved. We start by extending what it means to be a problematic overlap string to the many-object case:

**Definition 17** *(problematic overlap, general case) A window of length $L$ contains a* problematic overlap *if it contains no background and there is at least one overlap between two distinct objects inside the middle $L/2$ pixels. We call an $L$-pixel string a* problematic overlap string *if the only ways to generate it are via problematic overlaps.*

Next, we show that *any* problematic overlap string of sufficient length must have at least two objects in the same place as a reference version of that string (Lemma 33). Given this, we reason about a sufficient event for seeing such a problematic overlap string in a fixed image in Claim 34. This, in turn, allows us to apply the same Chernoff analysis as before, albeit with a slightly different union bound this time, as there are more problematic overlap strings in this case, in order to get the stated sample complexity. Full proof in Appendix C.4.

## 4. Inference

In this section, we discuss inference, providing algorithms for three different settings. Recall the inference task: given the set of objects and given an image, we must determine which objects were placed in what order and where on the canvas to generate this image. We consider two cases. First, if the objects are arbitrary (i.e., not necessarily $w$-well structured), then there could be multiple ways to generate any given image; we here will seek to reconstruct it with the fewest number of objects (Section 4.1). In this case, we consider a noiseless image generation process. Second, when we *do* have that objects are $w$-well-structured, we can aim for the stronger goal of getting the *correct* explanation for all but a small number of pixels. Here, we consider both noiseless (for which $w$-well-structuredness suffices, Section 4.2) and noisy (for which we need $\epsilon$-strongly, $w$-well-structuredness, Section 4.3) image generation. The problem of inference in these settings mirrors that of segmentation, studied in computer vision: for each pixel, we must determine its source object. Where the learning task corresponded to learning dictionary atoms, the inference task corresponds to finding "weights" for how the atoms are combined to give rise to a given instance.

### 4.1. Arbitrary Objects

Here, we consider the setting where the objects are (1) all of the same size, $s$, (2) known, and (3) arbitrarily generated, and our goal is to find the fewest number of objects required to generate the image. Formally, we want to find $S$ solving the problem below:

$$\min |S| \quad \text{s.t.} \quad \texttt{view(scene}(S)) = I \,.$$

Observe that there is a trivial solution that takes time $O(d^k \cdot m^k)$: simply enumerate the different images possible by placing one object, or two objects, or three objects, etc., into the canvas (or extended canvas for open room) until the first match is found. However, if $k \neq O(1)$, then this is computationally prohibitive. We provide a dynamic programming algorithm to solve the inference task in this setting in time $O(d\,s^2\,m^2)$. Details can be found in Appendix D.1.

### 4.2. Well-Structured Objects

In this section, we consider our original image-generation model and objects that satisfy $w$-well-structuredness. For inference in this model, we describe a greedy algorithm that considers identifiable objects in the image from objects with the most visible bits to those with the fewest visible bits and thereby successfully recovers a correct explanation for most of the bits of the image. While this algorithm makes more assumptions on the objects than the DP algorithm of Section 4.1 (namely, it assumes well-structuredness of objects, though it does not assume they are all the same size), it has several advantages. For instance, it will allow us to gain a tolerance to noise that we will show in Section 4.3. In addition, the assumption of well-structuredness allows us to discuss correctness, rather than minimality of an explanation. The background can be either distinct or well-structured. For a set $S$ as defined in Definition 11, we define $\hat{I} := \texttt{view(scene}(S))$. Formally, we solve the following problem where $d(\cdot, \cdot)$ is the Hamming distance: $\min_S d(\hat{I}, I)$.

Note that if there are $k$ objects in the image, there are at most $2k$ "transition" points, where a transition occurs when an object begins or ends. This means that in an image there are at most $2k+1$ *pure segments*, where each pure segment is a contiguous block of pixels arising from a single object or background. Our algorithm recovers an explanation for the image that for every pixel in the image will either correctly assign it to the object (or background) that generated it or else will output "I don't know", and moreover outputs "I don't know" on at most $4k^2w + 2k\,w$ locations.

**Brief Description of Algorithm**   The algorithm first scans for large chunks (at least $2k\,w + 1$) in the image that match a single object ("signatures") and chooses the largest one. Provided the image is larger than $(2k + 1) \cdot (2k\,w) + 1$, by the Pigeonhole principle, at least one such signature, for either an object or the background, must exist. Once the largest signature is identified, it is used to explain most of those pixels, and then the same algorithm is run recursively on the rest of the image. We keep running this algorithm on pieces that result until all pieces are too small to find a signature.

**Theorem 18**   *Whether the image has a distinct background (Defn. 4) or a well-structured background Defn. 5), the algorithm given in Algorithm 4 recovers an explanation for the image that is correct and explains at least $d - \Theta(wk^2)$ pixels and runs in polynomial time.*

**Proof Overview**   The main thing we argue is that the only way for any indices $i_{\text{start}}$ to $i_{\text{end}}$ to match a single object is if the region $i_{\text{start}} + k\,w$ to $i_{\text{end}} - k\,w$ arose from that object. Details are in Appendix D.2.

---

**Algorithm 4** Inference – ($\epsilon$-) Well-Structured Objects, (Noisy Copies), Greedy Algorithm

---

1: **Input:** $d$-pixel image, $\perp$, a character not in the image
2: explanation = { }
3: **while** parts of the image remain unexplained **do**
4:    $i_{\text{start}}, i_{\text{end}} \leftarrow$ largest string that ($\alpha$-approximately) matches a single object or background
5:    **if** $i_{\text{end}} - i_{\text{start}} \le 2\,k\,w$ **then**
6:      stop
7:    **else**
8:      associate $i_{\text{start}} + k\,w$ to $i_{\text{end}} - k\,w$ with the object
9:    **end if**
10:   Add the segment along with this object and relevant indices to explanation
11:   Replace explained bits with $\perp$
12: **end while**
13: **return** explanation

---

### 4.3. Noisy Image Generation

In this section, we explore what happens when the image generation process is noisy. We ask: if an adversary can corrupt some number of pixels of the image, do we have any hope of solving the segmentation problem? We first give an example that show how badly segmentation can go if an algorithm seeks exact matches but gets noisy images. Then, we provide an algorithm that completes the segmentation process and can correct noisy pixels under some assumptions.

**Example where exact match fails**   Suppose we are in the open room model. We have two colors and well-structured objects $A$ and $B$ (each length $d$), where object $A$ starts with the first color and object $B$ with the second color. In order to generate the image, pick one of the objects and place it on the canvas so it is fully visible. Next, pick a location $i$ in the middle $d/2$ pixels of the image and flip the bit in that location. This completes the adversarially-corrupted image. We can show that we require at least $d/(4w)$ objects to exactly explain this image, when in reality one object with one corrupted pixel suffices. More details are in Appendix D.3.1.

**Solving The Problem**   We assume objects are $\boldsymbol{\epsilon}$-strongly, $w$-well-structured. Further, if any window of length $W$ in the image can have up to $\boldsymbol{\alpha}$-fraction of its pixels corrupted, we are able to use a similar algorithm to before for recovery. We explore the regime where these small changes could significantly confuse an inference algorithm. We require that all the objects are substantially bigger than both the size of the signature and the size of segments the adversary can act on. The output of Algorithm 4 with the modifications in color provides an explanation that specifies the *correct* version of the image at those pixels, allowing us to also identify *where* the adversarial corruption occurred.

**Definition 19**  *We say that an adversary acting on an image $\mathcal{I}$ has* strength $(\alpha, W)$ *if the adversary can corrupt up to $\alpha$ fraction of the pixels in any window of size $W$ in $\mathcal{I}$. We call the image after corruption $\tilde{\mathcal{I}}$.*

    With this definition and objects that are $\epsilon$-strongly, $w$-well-structured, we can use an algorithm quite similar to Algorithm 4 to give an explanation for the image in terms of the objects, provided $\alpha$ and $\epsilon$ are appropriately related and the algorithm considers appropriately sized segments.

We require $\alpha < \epsilon/4$ because otherwise a corrupted segment of size $\max\{w, W\}$ could be closer to a piece that is not its source (by triangle inequality; details in Lemma 43). When considering adversaries as defined in Definition 19, our algorithm must use pieces large enough to ensure that even after this kind of corruption, the piece can be uniquely identified with a single object. For this, the algorithm must use $w_{\text{alg}} := \max\{w, W\}$, where $w$ is the well-structuredness parameter. Now, the algorithm proceeds as before, except that it considers *approximate* matches.

**Definition 20** *A string $\sigma$ from an image $\alpha$-approximately matches a string $\sigma^\star$ from an object if every window of length $w_{alg}$ in $\sigma$ requires at most $\alpha$ fraction of its pixels changed to exactly match $\sigma^\star$ in the corresponding pixels, i.e. for every $i$, $d(\sigma[i : i + w_{alg} - 1], \sigma^\star[i : i + w_{alg} - 1]) \le \alpha\, w_{alg}$, where $d(\cdot, \cdot)$ is the Hamming distance between the two strings.*

**Theorem 21** *Suppose an image is generated from $\epsilon$-strongly, $w$-well-structured objects, and an adversary of strength $(\alpha, W)$, $\alpha < \epsilon/4$, acts on it. Then, Algorithm 4 with the modifications in parentheses would recover in polynomial time an explanation for the unadulterated image that is correct in $d - \Theta(w_{alg}k^2)$ pixels.*

**Proof Overview** Our proof overall follows the same argument as before, with a couple added subtleties. Signatures are different in this setting, since we must check for chunks of the image that are *approximate matches* to the objects, i.e., if every contiguous block of $w_{\text{alg}}$ matches up to $\epsilon$ fraction of the pixels. Details can be found in Appendix D.3.

## 5. Discussion

In this work, we considered questions surrounding decomposition of complex but structured "images" into simpler, latent "objects," via some combinatorial process. We started by defining the setting we consider, including the property of being $w$-well-structured (similar to incoherence assumptions in standard dictionary learning). This property is a natural one to consider, as it holds with high probability for random, and even semi-random, objects. This property is powerful for both learning objects from multiple images and segmenting images correctly into the relevant objects.

We then studied two problems, "learning" and "inference": the learning problem asked how to decompose a set of complex items that derive their structure from a combinatorial process over latent components into a set of simple components. The inference problem asked, given the set of latent objects, how to decompose a single complex item into its component objects under various objectives: minimizing the number of objects required to explain and being confident of the source of each pixel both with and without adversarial corruptions. In the lattermost case, our work is a setting where the property of "having a simple explanation" can help protect against adversaries.

Some limitations of this work are the fact that we only study images and objects with one linear dimension, and we only study quantized colors. Also, one could study broader models of the joint distribution over the subset of objects that appears in the image. An additional model in which to study these questions is one where instead of placing objects directly on the canvas, we have the freedom to chop off the some parts of the ends, in a form of collaging. Elaborating on these factors would be interesting directions for future work.

Even in one dimension, another interesting direction is to consider is complicating the model by allowing different images to have different "views" or "lighting conditions" of any object. Now,

we can no longer look for identical pieces. Instead, suppose we have an oracle that tells us whether two strings arise from different views of the same underlying string or not. One complication that arises is that a particular string could match two different strings under two different decodings, and in this case, it is not immediately clear how to decode pieces consistently in order to ensure that the pieces can be sequenced. An important question is how to formalize this kind of variation and solve the problem.

## Acknowledgments

This work was supported in part by the National Science Foundation under grant CCF-1815011, by the NSF-Simons Funded Collaboration on the Mathematics of Deep Learning, and by the Defense Advanced Research Projects Agency under cooperative agreement HR00112020003. The views expressed in this work do not necessarily reflect the position or the policy of the Government and no official endorsement should be inferred.

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

## Appendix A. Related Work

**Related Work**  The issue of finding a small number of explanatory features for high-dimensional data is well-studied. Some particular framings of this problem include matrix factorization, for which principal components analysis (PCA) is often used (Karl Pearson, 1901; Hubert et al., 2000), dictionary learning, framed slightly more abstractly, (Kreutz-Delgado et al., 2003), and factor analysis (Harman, 1960). While much of the work traditionally done in this area focuses on the linear case, there have been several works that address non-linear settings: Mairal et al. (2009) consider a supervised dictionary learning problem which they solve by minimizing a risk function; Balcan et al. (2015, 2020) study representations for lifelong learning and exploit shared structure in the learning tasks to develop good summarized representations; Papyan et al. (2017) study a convolutional version of the problem.

Several works have also studied the use of decomposing high-dimensional objects into low-dimensional ones in the context of adversarial robustness (Bhagoji et al., 2018; Gupte et al., 2022). Most theory regarding adversarial robustness in computer vision considers the problem of classification (Cullina et al., 2018; Montasser et al., 2020, 2022). While this is an important setting to look at, much practical interest is in segmentation (Hendrik Metzen et al., 2017; Yatsura et al., 2022; Xie et al., 2017).

Another line of work related to this is that of factorial learning, studied extensively in the early 2000s. In this problem, any given object has a fixed ground truth appearance in 3d, but when placed into an image, there are finitely many ways it might be visible based on lighting, angle, etc. Thus, a fixed set of images are each available in an image, albeit in different configurations in different images. The goal, then, given an image, is to determine the parameters that dictate in which configuration each object in the image is. A common method to solve for the parameters is to solve an maximum likelihood problem using expectation maximization (Ghahramani, 1994).

A similar formulation is that of layered models, which decompose images into layers based on depth in order to then segment the image; here, too, the method is to model the image as a graphical model and then run an inference algorithm (Wang and Adelson, 1994; Jojic and Frey, 2001; Yi Yang et al., 2012). While this problem shares some characteristics with ours (learning what is in an image from samples), in our work we focus on a setting where there are no rotations, camera angles, or perspective shifts, and we instead focus on learning the set of objects from samples and then determining whether and where the objects are in a new image.

Others have explored learning objects from images in an unsupervised manner. Titsias (2005) does this by creating a probabilistic generative model that accounts for background generation, object placement, and any translations of the objects. They provide a greedy algorithm proceeding by "layer" (depth of object) that finds one object at a time. They then extend it to learning multiple objects in video data. This greedy algorithm learns the parameters of the aforementioned probabilistic graphical model. Their work focuses on learning objects as best as possible in realistic settings, while our work focuses on a controlled setting in which we can exactly learn the objects.

Another area of work on which we draw is computational biology. A well-studied problem in gene sequencing is reconstructing a string based on having seen recurring pieces of it (Motahari et al., 2013). We adapt such a shotgun sequencing algorithm for our use, as it will be applicable throughout Section 3. An extension we mention in Section 5, the collage model, can also be motivated by computational biology, where rather than having objects obscuring those they are in front of, we have strings (DNA segment) from pieces of other strings (genes).

## Appendix B. Details of Preliminaries

### B.1. Details of Models

In the following table, we summarize the places in which each of the models is considered in the paper.

| Section | Object Selection | Room Style | Depth | # objects in image | endpoint markers |
|---------|------------------|------------|-------|--------------------|------------------|
| Section 3.1 | uniform | open room | fully random | up to $m$ | yes |
| Section 3.1 | uniform | open room | partially-random | $k$ | yes |
| Section 3.2 | uniform | open room | partially-random | 2 | no |
| Section 3.3 | uniform | open room | partially-random | $k$ | no |
| Section 4.1 | anything | anything | anything | up to $m$ | no |
| Section 4.2 | anything | anything | anything | $k$ | no |
| Section 4.3 | anything | anything | anything | $k$ | no |

### B.2. Random and Semi-random Objects Are Well-Structured

In this section, we provide a proof for Lemma 3 and the state and prove Lemma 22, a similar result albeit for semi-random objects.

**Lemma 3** *[Random Objects are $w$-Well-Structured whp] A set of $m$ objects, each sampled uniformly at random from $\{0, 1, \ldots, c - 1\}^{s_i}$, and $s := \max_i s_i$ is $w$-well-structured with probability at least $1 - 3m^2 s^2 / c^w$. In particular, $w = O(\log ms)$ is sufficient so that the $m$ objects are $w$-well-structured with probability $1 - o(1)$.*

**Proof** In order to show this, we consider three cases:

1. **Windows in different objects:** Consider a window of size $w$ in one object and compare it to a window of size $w$ in a different object. There is a $1/c^w$ probability that they are the same. There are at most $\binom{m}{2} \cdot s^2$ pairs of indices in this category. So the probability of collision here is at most $m^2 s^2 / c^w$.

2. **Disjoint windows in a single object:** For two disjoint windows of size $w$ in the same object, the same holds, and the probability that they are the same is $1/c^w$. There are at most $m \cdot s(s - w)$ pairs of indices in this category, so the probability of collision is at most $m \cdot s^2 / c^w$.

3. **Overlapping windows in a single object:** now for overlapping windows in a single object, we consider the strings location-by-location. Suppose the offset is $q \in \{1, \ldots w - 1\}$. Then within object $A$, say we are interested in the probability that $A[i : i + w - 1] = A[i + q : i + q + w - 1]$. This is equivalent to the intersection of events that $A[i] = A[i + q]$, $A[i + 1] = A[i + 1 + q], \ldots A[i + w] = A[i + q + w]$. Since pixels of the object are generated independently at random, the probability here is also $1/c^w$ (in particular, if one considers the locations in $A$ being generated left to right, then $A[i + q]$ has a $1/c$ chance of being equal to $A[i]$, and conditioned on that $A[i + 1 + q]$ has a $1/c$ chance of being equal to $A[i + 1]$, and so on). There are at most $m \cdot s \cdot w$ pairs of indices in this case, so the probability of collision is at most $m \cdot s^2 / c^w$.

Thus, adding these three cases together, the probability that two $w$-character strings are the same for randomly-generated objects is at most:

$$\frac{m^2 s^2}{c^w} + \frac{ms^2}{c^w} + \frac{ms^2}{c^w} \leq \frac{3m^2 s^2}{c^w} .$$

∎

**Lemma 22** *[Semi-random Objects are $w$-Well-Structured whp] Suppose an adversary determines what the objects look like. After all the adversary's decisions are made, for each location in each object, the color in that location is replaced by a random color with probability $p$. Then, a set of $m$ objects over $\{0,1\}^s$ generated as described in this way is $w$-well-structured with probability at least $1 - 3m^2 s^2 (1 - p(1 - 1/c))^w$. In particular, $w = O(\frac{1}{p} \log(ms))$ suffices for the $m$ objects to be $w$-well-structured with probability $1 - o(1)$.*

**Proof**

As before, we will compute the three cases: different objects, same object no overlap, same object overlap. Now we focus on case 3 because the first two cases are easier due to independence. We think about proceeding by tossing coins left to right and fixing colors to the left of the current location. In this case, the probability that the color in the $i + w^{\text{th}}$ location is different from color in the (fixed) $i^{\text{th}}$ location is at least the probability that we change it from the original, $p$, multiplied by the probability of landing on a different color, $1 - 1/c \geq 1/2$. Thus, the probability that colors in these two locations are the same is at most:

$$q := 1 - \frac{p}{2} .$$

Thus, the probability of two locations have the same color is at most $q := 1 - \frac{p}{2}$.

Now, we can consider the same three cases as above, and the same logic applies, though now with probability $q^w$ rather than $c^{-w}$. Then, the probability of $w$-pixel strings being the same is at most:

$$m^2 s^2 q^w + m \cdot s^2 q^w + m \cdot s^2 q^w \leq 3m^2 s^2 q^w \leq 3m^2 s^2 (1 - p(1 - p))^w \leq 3m^2 s^2\, e^{-\frac{p}{2} w} .$$

Thus, for the probability of two $w$ length strings to be small, $w = O(\log(m\,s)/p)$

∎

**B.3. Random Objects Are $\epsilon$-strongly Well-Structured**

**Lemma 23** *A set of $m$ objects, each sampled uniformly at random from $\{0, 1, \ldots c-1\}^{s_i}$, and $s := \max_i s_i$ is $\epsilon$-strongly, $w$-well structured with probability at least $1 - 3m^2 s^2\, e^{w((1-\epsilon) - (1-\epsilon)\ln(1-\epsilon)c - 1/c)}$. For example, if $c = 2$ and $\epsilon = 1/10$ then the set will be $\epsilon$-strongly $w$-well-structured with probability at least $1 - 3m^2 s^2 e^{-0.129w}$; for $c = 4$ and $\epsilon = 1/10$ the probability is at least $1 - 3m^2 s^2 e^{-0.503w}$. In particular, $w = O(\frac{-1}{(1-\epsilon) - (1-\epsilon)\ln((1-\epsilon)c) - 1/c} \log ms)$ is sufficient so that the $m$ objects are $\epsilon$-strongly, $w$-well-structured with probability $1 - o(1)$.*

**Proof** As before, we must consider the same three cases. In each case, now, instead of considering the probability that each pixel of the string matches, we consider the probability that more than a $1-\epsilon$ fraction of the pixels match. For this, let us first let $X_i = \mathbb{I}\{\text{pixel } i \text{ of string 1 matches pixel } i \text{ of string 2}\}$. Further, let $X = \sum_i X_i$. Now, to argue that too many pixels don't match, we consider what fraction of pixels we expect to see matches in and then argue using tail bounds that with high probability, we would not see more than $(1 - \epsilon)\,w$ of the pixels matching. To start, we formally state the tail bound that we use, adapted from Theorem 4.4, Part 1 in Mitzenmacher and Upfal (2005):

**Lemma 24 (Large Deviation Chernoff)** *Let $X_1, X_2, \ldots, X_n$ be independent Poisson trials such that $\mathbb{P}[X_i] = p_i$. Let $X = \sum_{i=1}^{n} X_i$ and $\mu = \mathbb{E}[X]$. The following Chernoff bound holds: for any $\delta > 0$,*

$$\mathbb{P}[X \geq (1 + \delta)\mu] \leq \left( \frac{e^\delta}{(1 + \delta)^{(1+\delta)}} \right)^\mu$$

Since the colors are chosen independently at random, in expectation, in a $w$-pixel string we expect $w/c$ pixels to match. That is, $\mathbb{E}[X] = w/c$. We wish to analyze the probability that more than $(1 - \epsilon)\,w$ pixels match. From the Large Deviation Chernoff bound, setting $\delta := (1 - \epsilon)c - 1$, we have that where $\delta > 0$, i.e., $1 - \epsilon > 1/c$:

$$\mathbb{P}\left[ X \geq (1 + \delta)\frac{w}{c} \right] \leq \left( \frac{e^{(1-\epsilon)c-1}}{((1 - \epsilon)\,c)^{(1-\epsilon)\,c}} \right)^{\frac{w}{c}} = e^{((1-\epsilon)w - w/c)} \left( \frac{1}{(1 - \epsilon)\,c} \right)^{(1-\epsilon)\,w} \quad (1)$$

$$= e^{(1-\epsilon)w - w/c} e^{-(1-\epsilon)w \ln((1-\epsilon)c)} \quad (2)$$

As before, we take the union bound over at most $3m^2 s^2$ strings, so the probability that *any* of these strings fails is at most $3m^2 s^2\, e^{(1-\epsilon)w - w/c} e^{-(1-\epsilon)w \ln((1-\epsilon)c)}$.

$$\frac{d}{dw}\left( (1 - \epsilon)w - w/c - (1 - \epsilon)w \ln((1 - \epsilon)c) \right) = (1 - \epsilon) - \frac{1}{c} - (1 - \epsilon) \ln\left((1 - \epsilon)c\right) \quad (3)$$

All that remains to be checked, then, is that in the regime of interest, this quantity is negative.

To do this, let us take the derivative with respect to $\epsilon$, to show that for a fixed value of $c$, for any value of $\epsilon$, this quantity is negative

$$-1 - \left( -\frac{1 - \epsilon}{(1 - \epsilon)c} + -1 \ln((1 - \epsilon)c) \right) = -1 + \frac{1}{c} - \ln((1 - \epsilon)c) < 0. \quad (4)$$

Finally, we confirm that the starting point is non-positive. When $1 - \epsilon = 1/c$, the derivative in Eqn. 3 is 0. Thus, the exponent is negative and the probability is decreasing with increasing $w$.

Thus, we have that if $w$ if sufficiently large, the failure probability can be very small. In particular, if $w = O(\frac{-1}{(1-\epsilon)-(1-\epsilon)\ln((1-\epsilon)c)-1/c} \log ms)$, then the failure probability is $o(1)$. ∎

## Appendix C. Proofs for Learning Algorithms

### C.1. Proof of Lemma 12

**Lemma 12** *In the open room, uniform object selection, partially-random model, the probability of seeing a given L-pixel segment of a given object (or even of an $(L-1)$-padding of that object) in a random image composed of $k$ objects is at least $\left(1 - \frac{s+L-1}{d'}\right)^{k-1} \cdot \frac{d+1-L}{d'} \cdot \frac{k}{m}$.*

**Proof** There are three pieces to consider: (1) whether the object shows up in the image; (2) whether the desired piece of the object shows up in the image given (1); (3) whether the desired section remains unobscured given (1) and (2). Suppose the largest object in the set has size $s$. In the uniform object selection model, the probability the object shows up in the image is $\frac{k}{m}$. The probability of a fixed length-$L$ piece (of the object or of a padding of the object) appearing on the canvas given the object appears in the image is at least $\frac{d+1-L}{d'}$. The probability of a fixed length-$L$ string being obscured by an object that appears in front of it is at most $(s+L)/d'$. Since objects are placed independently, even if the object of interest is in the back, the probability that none of the $k-1$ other objects placed on the canvas obscures this is at least $\left(1 - \frac{s+L-1}{d'}\right)^{k-1}$. Thus, the probability that the full $L$-pixel piece is visible in the image is:

$$\mathbb{P}\left[\text{no other objects obscure it} \,\middle|\, \text{desired object and section appear}\right] \cdot \tag{5}$$

$$\mathbb{P}\left[\text{desired section visible} \,\middle|\, \text{object in image}\right] \cdot \mathbb{P}\left[\text{this object is in image}\right] \tag{6}$$

$$\geq \left(1 - \frac{s+L-1}{d'}\right)^{k-1} \cdot \frac{d+1-L}{d'} \cdot \frac{k}{m} =: a. \tag{7}$$

Padding is not relevant in this section, but it will become important when we no longer have endpoint markers and therefore need to discard parts of strings to be confident they belong to a single object (Section 3.2).

■

### C.2. Several Objects per Image with Endpoint Makers (Theorem 13)

**Proof**

At a high level, we show this theorem by first computing the probability of seeing a fixed $L$-pixel string from an object in a fixed image. Then, we compute the probability of seeing it in any image and finally, take a union bound.

First, we recall Lemma 12, in which we compute a lower bound on the probability of seeing a fixed $L$-pixel string in an image.

Then, as before, let $a = \left(1 - \frac{s+L-1}{d'}\right)^{k-1} \cdot \frac{d+1-L}{d'} \cdot \frac{k}{m}$ be the probability bound in Lemma 12. The probability a given $L$-pixel string of a given object is visible in at least one of $S$ independently generated images is, therefore, at least $1 - (1-a)^S$. The total number of pieces per object we need to see is at most $\frac{2s}{L}$, since we need to see pieces with enough overlap to align them (Figure 1). Then,

the probability that at least one of the required pieces is obscured is given by the union bound:

$$\mathbb{P}\left[\text{not seeing at least one of the required pieces in any of the images}\right] \leq ((1-a)^S)\frac{2\,m\,s}{L} \tag{8}$$

$$\Leftrightarrow \mathbb{P}\left[\text{seeing all required pieces}\right] \geq 1 - ((1-a)^S)\frac{2\,m\,s}{L} \tag{9}$$

Next, to ensure this probability is small enough:

$$\delta = ((1-a)^S)\frac{2\,m\,s}{L} \leq e^{-aS}\frac{2\,m\,s}{L} \tag{10}$$

$$\text{want this to be} \qquad \leq \frac{1}{10} \tag{11}$$

$$\Leftrightarrow e^{-aS} \leq \frac{L}{20\,m\,s} \tag{12}$$

$$\Leftrightarrow \frac{20m\,s}{L} \leq e^{aS} \tag{13}$$

$$S \geq \frac{\ln(20\,m\,s/L)}{a} \tag{14}$$

$$\tag{15}$$

Thus, we require $S = \Theta\left(\frac{\ln(ms/L)}{a}\right)$ to see all the pieces required with high enough probability.

∎

### C.2.1. EXTENSION

This result would also extend to the case where objects are $\epsilon, w$-strongly-well-structured, and at most $\epsilon/2$ fraction of the pixels are corrupted by an adversary before an object is placed. In that case, when seeing an $L$-pixel piece, we would have identify what the uncorrupted version of it was before feeding it into the sequencing algorithm. Since at most $\epsilon/2$ fraction of pixels in an $L$-pixel string are corrupted, this string is closer to the respective $L$ pixel string from its source object than any other object (see Lemma 43 for formal argument regarding this fact).

### C.3. Two Objects per Image Case

In this section, we prove Theorem 15. To do so, we show that for an appropriate value of $L$, length-$L$ substrings of a single object will appear significantly more frequently in images than length-$L$ strings created by overlaps of multiple objects, and therefore we can use frequency of occurrence to identify strings to be stitched together to reconstruct the objects. We first provide the building blocks of this argument and then show how to combine them to prove the theorem.

We first bound the probability of seeing a fixed length-$L$ segment in a random image. This follows from Lemma 12.

**Claim 25** *Under the open room, uniform, partially-random model, the probability of seeing a given $L$-pixel segment of a single object in a random image composed of two objects, is at least $\left(1 - \frac{s+L-1}{d'}\right) \cdot \frac{d+1-L}{d'} \cdot \frac{2}{m}$ and at most $\frac{d+1-L}{d'} \cdot \frac{2}{m}$.*

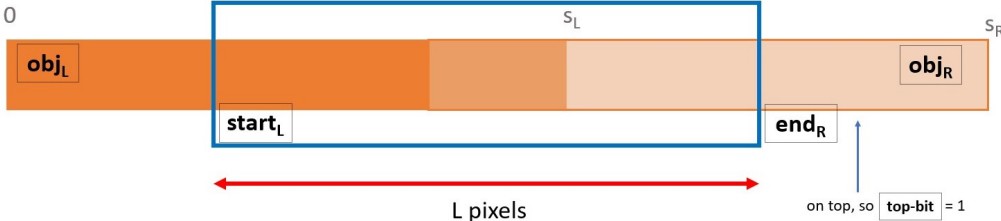

Figure 3: Consider the $L$-pixel problematic overlap string boxed in blue. There are 5 different features that uniquely define it. The first is identity of the left object (darker orange), followed by the index within it where the string of interest (boxed in blue) starts. Third, we care about the identity of the right object (lighter orange) and the index within it at which the string of interest ends. Note that these uniquely identify how much overlap there must be, since the length $L$ is fixed. Finally, we include a bit to make clear which object is on top.

**Proof** As in Lemma 12, the probability the desired segment is visible is the probability the object is in the image, times the probability the desired segment appears in the window given that the object is in the image, times the probability no other objects obscure the segment given that the desired object and segment appear. This is:

$$\begin{cases} \leq 1 \cdot \frac{d+1-L}{d'} \cdot \frac{2}{m} & \text{maximized if object is always in the front when it appears,} \\ \geq \left(1 - \frac{s+L-1}{d'}\right) \cdot \frac{d+1-L}{d'} \cdot \frac{2}{m} & \text{minimized if object is always in the back when it appears .} \end{cases}$$

∎

Recall that a problematic overlap is a window of size $L$ that contains two objects that overlap inside the middle $L/2$ pixels, and no background. We call an $L$-pixel string a *problematic overlap string* if the only ways to generate it are via problematic overlaps. We show that problematic overlap strings that look identical must be (almost) identical. In particular, the only thing that could differ between them is which object is on top. Consider the following tuple-based description of an $L$-pixel string: $(\text{OBJ}_L, \text{START}_L, \text{OBJ}_R, \text{END}_R, \text{TOP-BIT})$ (see Figure 3).

**Lemma 26** *For a set of objects that are $w$-well-structured, an $L$-pixel problematic overlap string can be generated via at most two different five-tuples as described above, provided $L \geq 4w$. In particular, any two five-tuples that generate the same problematic overlap string must agree in $\text{OBJ}_L, \text{START}_L, \text{OBJ}_R, \text{END}_R$ .*

**Proof** Suppose, for contradiction, that they differ in the $\text{OBJ}_L$. That is, there are two different objects, $\text{OBJ}_{L1}$ and $\text{OBJ}_{L2}$ that can produce that problematic overlap. Then, the leftmost $L/4 = w$ pixels must match between them, which is a violation of the well-structredness property. Likewise, the $\text{OBJ}_R$ cannot differ. Finally, if the objects remain the same but one or both of $\text{START}_L, \text{END}_R$

differ, then an object must have $L/4 = w$ pixels that are the same under shift. This, too, is a violation of well-structuredness. Thus, the only possible difference is if the object on top changes, in which case some small number of pixels would be shared between the end of the left object and the start of the right object. With this, we have shown the Lemma.

∎

Now that we know that there are certain fixed events that must occur for a particular problematic overlap string to show up, we bound the probability of seeing such a string based on the probability of those events.

**Claim 27** *Under the open room, uniform, partially-random model with $w$-well-structured objects, where $w \leq L/4$, the probability, $p_{bad}$, that a particular problematic overlap string appears is at most:*

$$\frac{2}{\binom{m}{2}} \cdot \frac{d - L}{d'^2}$$

*and at least:*

$$\frac{1}{\binom{m}{2}} \cdot \frac{d - L}{d'^2}.$$

**Proof** Let $s$ be a particular problematic overlap string. From Lemma 26, there must be two objects $A$ and $B$ that comprise it, which must be placed at a specific location relative to each other. The lemma also tells us that either only one ordering of them works to generate it, or both do. Based on this:

$$p_{\text{bad}} := \mathbb{P}\left[A, B \text{ in object}\right] \cdot \mathbb{P}\left[A, B \text{ placed correctly} \,\middle|\, A, B \text{ in image}\right] \cdot (\text{number of orderings that work}) \tag{16}$$

$$= \frac{1}{\binom{m}{2}} \cdot \frac{d - L}{d'^2} \cdot \begin{cases} 1 & \text{if only 1 ordering fine} \\ 2 & \text{if either ordering fine} \end{cases} \tag{17}$$

∎

We can show that the probability of seeing good strings and that of seeing bad strings is adequately separated:

**Claim 28** *If $s + 2L < 3d/2$ and $m\,d' \geq 128$, then $p_{good}$ is separated from $p_{bad}$, with $p_{mid} := \frac{1}{16\,m}$ such that $p_{good} > 2\,p_{mid}$ and $p_{mid} > 2 \cdot p_{bad}$.*

**Proof** We show first that $p_{\text{good}} \geq 2p_{\text{mid}}$. From Claim 25:

$$\frac{p_{\text{good}}}{2} \geq \left(1 - \frac{s + L - 1}{d'}\right) \cdot \frac{d + 1 - L}{d'} \cdot \frac{2}{m} \cdot \frac{1}{2} = \frac{(d' - (s + L - 1)) \cdot (d + 1 - L)}{d'^2} \cdot \frac{1}{m} \tag{18}$$

$$= \frac{(d + s - L - 1)(d + 1 - L)}{(d + s - 2)^2} \cdot \frac{1}{m} \geq \frac{(d + 1 - L)^2}{(d + s - 2)^2} \cdot \frac{1}{m} \tag{19}$$

$$= \left(1 - \frac{s - 3 + L}{d + s - 2}\right)^2 \cdot \frac{1}{m} \tag{20}$$

$$\geq p_{\text{mid}} \qquad \text{if} \qquad \frac{s + L}{d + s - 2} < \frac{3}{4} \tag{21}$$

Next, we must show that $p_{\text{bad}} \leq \frac{p_{\text{mid}}}{2}$, for which it suffices to show that the upper bound on $p_{\text{bad}}$ from Claim 27 is bounded by this quantity.

$$2 \cdot p_{\text{bad}} \leq \frac{8}{m^2} \cdot \frac{d - L}{d'^2} = \frac{8}{m^2} \cdot \frac{d - L}{d'^2} \tag{22}$$

$$= \frac{1}{16\,m} \frac{128(d - L)}{m\,d'^2} \leq p_{\text{mid}} \ \text{ if } \ \frac{128(d - L)}{m\,d'^2} \leq 1 \ \ \Leftarrow m\,d' \geq 128. \tag{23}$$

Thus we complete the proof of this claim. ∎

Finally, we can use Chernoff's inequality to bound the probability of success, once we know the number of good strings, number of bad strings, and a probability $p_{\text{mid}}$ such that the probability of seeing a fixed good string is at least twice as large and the probability of seeing a bad string is at most half as much.

**Lemma 29 (Use Chernoff to Get Sample Complexity)** *Suppose there exists $p_{mid}$ such that $p_{bad} \leq p_{mid}/2$, and $p_{good} \geq 2p_{mid}$. If there are $n_{po}$ problematic overlap strings and $n_{single}$ strings of length $L$ comprised of a single object that must be seen to reconstruct all objects, then $S = O\left( \max\left\{ \frac{\ln(n_{single})}{p_{mid}}, \frac{\ln(n_{po})}{p_{mid}} \right\} \right)$ image samples suffice to see all the necessary strings for reconstructing the objects and discard problematic overlap strings with probability 9/10.*

**Proof** We compute the probabilities of seeing too few copies of a good string or too many copies of a bad string and control that. Recall that the probability of seeing a fixed good string in a given image is $p_{\text{good}}$ and an upper bound on the probability of seeing a fixed bad string in a given image is $p_{\text{bad}}$. Fix a good string $\sigma$ of length $L$. If we draw $S$ samples, in expectation, we should see $p_{\text{good}} \cdot S$ copies of $\sigma$. For sample $j$, let $X_j = \mathbb{I}(\sigma \text{ in sample } j)$. Since the images are independent, we may use Chernoff bound:

$$\mathbb{P}\left[ \sum_{j=1}^{S} X_j \leq \left(1 - \frac{1}{2}\right) p_{\text{good}} \cdot S \right] \leq \exp\left( \frac{-p_{\text{good}} \cdot S}{8} \right) \tag{24}$$

$$\leq \exp\left( \frac{-p_{\text{mid}} \cdot S}{4} \right) \tag{25}$$

Similarly, for a bad string $\sigma'$, we expect to see $p_{\text{bad}} \cdot S$ copies of it. For sample $j$, let $Y_j = \mathbb{I}(\sigma' \text{ in sample } j)$. Since the images are independent, we may use Chernoff bound:

$$\mathbb{P}\left[ \sum_{j=1}^{S} Y_j \geq (1 + 1) p_{\text{bad}} \cdot S \right] \leq \mathbb{P}\left[ \sum_{j=1}^{S} Y_j \geq (1 + 1) \frac{p_{\text{mid}}}{2} \cdot S \right] \tag{26}$$

$$< \left( \frac{e}{4} \right)^{\frac{p_{\text{mid}} \cdot S}{2}}. \tag{27}$$

Now that we know the probabilities of seeing a fixed string sufficiently many times, we wish to now union bound over the respective $n_{\text{single}}$ good and $n_{\text{po}}$ bad strings. The probability that any of the $n_{\text{single}}$ good strings does not appear at least $p_{\text{good}} S/2$ times is at most $n_{\text{single}} \cdot \exp\left( \frac{-p_{\text{mid}} \cdot S}{4} \right)$.

Similarly, the probability that any of the $n_{\text{po}}$ bad strings appears more than $2p_{\text{bad}}S$ times is at most: $n_{\text{po}}\left(\frac{e}{4}\right)^{\frac{p_{\text{mid}} \cdot S}{2}}$ .

In order to keep both of these probabilities small, we would require:

$$n_{\text{single}} \cdot \exp\left(\frac{-p_{\text{mid}} \cdot S}{4}\right) \leq \frac{1}{10} \quad \Leftrightarrow \quad \frac{-p_{\text{mid}} \cdot S}{4} \leq \ln\left(\frac{1}{10\, n_{\text{single}}}\right) \tag{28}$$

$$\Leftrightarrow \quad \frac{4}{p_{\text{mid}}}\ln\left(10\, n_{\text{single}}\right) \leq S \tag{29}$$

$$n_{\text{po}}\left(\frac{e}{4}\right)^{p_{\text{mid}} \cdot S/2} \leq \frac{1}{10} \quad \Leftrightarrow \quad \frac{p_{\text{mid}}}{2} \cdot S \cdot \ln(e/4) \leq \ln\left(\frac{1}{10\, n_{\text{po}}}\right) \tag{30}$$

$$\Leftrightarrow \quad \frac{2\ln(10 n_{\text{po}})}{p_{\text{mid}}\ln(4/e)} \leq S\,. \tag{31}$$

Thus, for some constant $c$, if $S \geq c \cdot \frac{\ln(n_{\text{single}})}{p_{\text{mid}}}$, then we recover all necessary pieces of objects correctly with high probability. Similarly, for some constant $c'$, if $S \geq c' \cdot \frac{\ln(n_{\text{po}})}{p_{\text{mid}}}$, then we don't see problematic overlap strings too frequently. From here, we can run the sequencing algorithm given in Algorithm 1 to recover the objects themselves accurately with high probability. Thus, $S = O\left(\max\left\{\frac{\ln(n_{\text{single}})}{p_{\text{mid}}}, \frac{\ln(n_{\text{po}})}{p_{\text{mid}}}\right\}\right)$ is sufficient for recovering the objects from sampled images. ■

With this, we are equipped to prove the theorem statement.

**Theorem 15** *In the open room, uniform, partially-random model with $w$-well-structured objects, each of length at most $s$ and at least $4w$, and 2 objects per image, Algorithm 3 with $L = 4w$ recovers all $m$ objects with $S = \Omega\left(m\ln(ms)\right)$ samples and $\tau = \frac{1}{16m}$ with probability $\frac{9}{10}$.*

**Proof**

The theorem follows from a direct application of Lemma 29 and argument that the ends are indeed recovered correctly. At a high level, we determine how many samples are required to see good strings frequently and see bad strings infrequently. Then, we argue that within such a set of samples, we have all we need in order for the end recovery subroutine to succeed. To apply Lemma 29, we need (1) $p_{\text{mid}}$, (2) the number of strings arising from a single object we must see, and (3) the number of problematic overlap strings which we need to not see too frequently. We have $p_{\text{mid}} = 1/(16m)$ from Claim 28.

Next, we compute the number of good and bad strings:

**Claim 30** *Over $m$ total objects, out of the $m \cdot (s - L + 1)$ good strings, there is a subset of size $4\, m\, s/L$ that is sufficient for reconstructing all of the objects.*

**Proof** From a single object, there are: $\frac{s}{w}$ distinct contiguous $L$-pixel portions that are sufficient for reconstructing the object. This is because we take $L = 4w$-pixel strings and only use the middle $L/2 = 2w$ pixels of them. As a result, we require overlap of $L/4 = w$ pixels in the used part, meaning that the shift between two pieces used is $L/4 = w$ (recall Figure 2). Then, there are $m$ objects for which we need all $s/w = 4s/L$ strings. ■

**Claim 31** *The total number of tuples generating problematic overlap strings is at most $\binom{m}{2} \cdot s^2 \cdot 2 \leq 2m^2 s^2$ .*

**Proof** The number of two-object $L$-pixel strings that could be problematic is at most the total number of tuples of two objects, two indices, and a bit to specify which is on top that could generate such a string. ■

Applying the result of Lemma 29 and incorporating Claims 28, 30, 31, we have that the number of required samples is:

$$O\left(\max\left\{\frac{\ln(n_{\text{single}})}{p_{\text{mid}}}, \frac{\ln(n_{\text{po}})}{p_{\text{mid}}}\right\}\right) = O\left(\max\left\{\frac{\ln(4ms/L)}{p_{\text{mid}}}, \frac{\ln(2\,m^2\,s^2)}{p_{\text{mid}}}\right\}\right) = O(m\,\ln(m\,s))\,.$$

Finally, we must show that the part of the algorithm that recovers the ends indeed does so correctly. From the analysis above, we have that the first two lines of Algorithm 3 must recover all of the objects except the leftmost and rightmost $L/4$ pixels of each. Thus, we must now argue that the `for` loop correctly recovers the ends of the objects. To do so for the left ends, we argue two things:

1. that the leftmost $w$ pixels of an object seen in sequencing (i.e., `sleftend`) must be seen at least once preceded by a segment with background and the true leftmost end of the object.

2. taking the shortest segment $x$ sandwiched between background and `sleftend` contains no problematic overlaps and therefore must be the true left end

The right ends follow symmetrically.

**Lemma 32** *The end recovery subroutine in Algorithm 3 correctly retrieves the ends of all objects.*

**Proof** Since we collect any string that appears at least $\tau$ fraction of the time and send it to sequencing, we in particular must have included pixels $L/4$ to $3L/4$ of the object. This is the furthest right possible for `sleftend`. From the padded version of Lemma 12, we know that we must also frequently see the string that is $L/4$ background, followed by $3L/4$ pixels of object in the set of images. This is furthest left possible. Thus there is sufficient overlap for us to be confident that we will see a segment that looks like background followed by the start of the object, up to and including the leftmost pixels seen in sequencing.

Now, we must show that the shortest segment $x$ that lies between background and the signature `sleftend` is indeed the true left end of the object. To this end, we observe that if we see background followed by non-background pixels, followed by a signature of an object, then that object must have started somewhere in the middle, and any pixels of other objects must also arise from the left ends of them. We immediately see, therefore, that no such string can be shorter than one where background is immediately followed by the object whose end we are seeking. Likewise, if other objects $o_i$ start in the middle, with $b_i > 0$ pixels of each visible, then the length of the segment between background and `sleftend` must be $\sum_i b_i$, which is longer than if only the object of interest is present. Thus, by selecting the shortest such segment, we are guaranteed to recover the true end of the object.

Finally, recovery of the right ends is exactly symmetric. ■

■

## C.4. Small Number of Objects per Image Case (Thm. 16)

In this section, we set out to prove Theorem 16 by deriving the number of samples needed to adequately reconstruct the objects. To do so, we show as before that for an appropriate value of $L$, length-$L$ substrings of a single object will appear more frequently in images than length-$L$ strings created by overlaps of multiple objects. In this section, the main difference from before is that these overlap strings can comprise more than just 2 objects. We first provide the building blocks of this argument and then show how to combine them to prove the theorem.

As before, we start by identifying what must occur each time we see a problematic overlap string that matches a reference string. Recall that there are at most $k$ objects per image and all of the objects are larger than $6wk$.

**Lemma 33** *Assume objects are $w$-well-structured. Suppose $L \geq 8\,w\,k$ and each object is of size at least $3L/4$. Consider a reference problematic overlap string of length $L$ pixels. Any other way of generating this string with $k$ objects must require that at least 2 objects are the same and in the same place as in the reference.*

**Proof** Consider a reference problematic overlap string of length $L$ that consists of at most $k$ object instances. We know that there must be a point in the middle $L/2$ positions where the current object changes, since it is a problematic overlap. We break the string up into the three regions: left (of size $L/4$), middle (of size $L/2$), and right (of size $L/4$), where the left and right regions are the ones we discard, and the middle is where the crossover occurs. Now, in the left region, we know all the pixels align between the reference string and its duplicate. Mark each time the current object changes on either the reference or the duplicate. Because objects have size greater than $L/4$, one that appears cannot disappear before the left region completes. Thus, there are at most $2k$ such changes, and so $2k + 1$ sections corresponding to these changes, in the left part of the string. The total number of pixels is $L/4$, so by Pigeonhole Principle, there is at least one object with at least $L/(8k + 4) \geq w$ pixels visible. Since our set of objects is $w$-well-structured, there is only one such object, and it must be in the same position. The exact same argument holds for the right end of the string. From this, we know that there are segments of at least length $w$ on both the left and right sides that match the reference.

Now, we show that these segments must come from separate object instances. Consider two cases: in case 1, if a new object starts in the middle, then as long as that new object has length $\geq 3L/4$, then the left one will not reappear. In case 2, the object on the left ended in the middle, so it does not account for the pixels on the right, and the object that showed up was not visible to the left of this. This shows the reference can't have same object on left and right. Then in the duplicate, if the left and right end objects are the same, it must have happened in the reference. Thus, since there is a crossover point in the middle, and the objects are all large enough, we know they must be different object instances. Thus, we have that at least two object instances must be in the same place regardless of the others in order to replicate a reference problematic overlap. ∎

We use this lemma to compute the probability of seeing such a problematic overlap string, based on considering the probabilities of the necessary events identified above.

**Claim 34** *Under the open room, uniform, partially-random model with $w$-well-structured objects, the probability $p_{bad}$ that a particular problematic overlap string appears is at most:*

$$\frac{k(k-1)}{2} \cdot \frac{d - L}{d'^2}.$$

**Proof**

From Lemma 33, we know that for every version of a problematic overlap string, there are two objects that *must* appear with a large number of pixels visible. Since every version has to agree with reference in at least 2 objects, we can partition the versions into at most $\binom{k}{2}$ categories, where each category corresponds to a specific pair of signature objects. If a version corresponds to multiple categories, we can arbitrarily assign it to one of them. For any fixed pair of signature objects, $A, B$, we can upper bound the probability $p_{A,B}$ that $A$ and $B$ appear and are placed correctly as follows:

$$p_{A,B} \leq \mathbb{P}\Big[ \text{two identifying objects appear and are placed correctly} \Big] \tag{32}$$

$$= \mathbb{P}\left[2 \text{ identifying objects are in image}\right] \cdot \mathbb{P}\left[\text{both are placed correctly} \,\Big|\, \text{both are in image}\right] \tag{33}$$

$$\leq 1 \cdot \frac{d - L}{d'^2} \tag{34}$$

$$\leq 1 \cdot \frac{d - L}{d'^2} \tag{35}$$

Multiplying this by $\binom{k}{2}$, in order to union bound over the different categories, we get that the total probability $p_{\text{bad}}$ is at most:

$$\frac{k(k - 1)}{2} \cdot \frac{d - L}{d'^2}$$

∎

Again, as before, we identify a $p_{\text{mid}}$ such that the probability of seeing good strings is at least twice $p_{\text{mid}}$ and the probability of seeing bad strings is at most half $p_{\text{mid}}$. Notably, now, $p_{\text{mid}}$ depends on the number of objects, since that affects how often the object is visible (when we can make no other assumptions about depth).

**Claim 35** *If $d > 2L$, $d' \geq 16m\,k\,2^k$ then $p_{good}$ is separated from $p_{bad}$, with $p_{mid} := k/(16\,m\,2^k)$, such that $p_{good} > 2p_{mid}$ and $p_{mid} > 2p_{bad}$.*

**Proof** First, let us consider $p_{\text{bad}}$. We have:

$$2 \cdot p_{\text{bad}} \leq 2 \cdot \frac{k^2}{2} \cdot \frac{d - L}{d'^2} \leq k^2 \cdot \frac{1}{d'} \tag{36}$$

$$\leq \frac{1}{16m} \frac{k}{2^k} \quad \text{provided } d' \geq 16\,m\,k\,2^k \tag{37}$$

$$= p_{\text{mid}} \tag{38}$$

Next, for the other side, we consider $p_{\text{good}}$, which is the probability computed in Lemma 12. Namely:

$$\frac{p_{\text{good}}}{2} \geq \frac{1}{2}\left(1 - \frac{s+L-1}{d'}\right)^{k-1} \cdot \frac{d+s-L}{d'} \cdot \frac{k}{m} \tag{39}$$

$$\geq \frac{1}{2}\left(\frac{d+s-L-1}{d'}\right)^{k} \cdot \frac{k}{m} \tag{40}$$

$$\geq \frac{1}{2}\left(\frac{1}{2}\right)^{k}\frac{k}{m} \quad \text{provided } \frac{d+s-L-1}{d'} \geq \frac{1}{2} \tag{41}$$

$$\geq \frac{1}{16m}\frac{k}{2^k} = p_{\text{mid}} \tag{42}$$

For $\frac{d+s-L-1}{d'} \geq \frac{1}{2}$, it suffices for $L < d/2$. Thus, there is a separation between the two with $p_{\text{mid}} := k/(16m\,2^k)$. ∎

With these pieces, we are ready to prove the following theorem, restated from before:

**Theorem 16** *In the open room, uniform, partially-random model with $w$-well-structured objects, each of length at most $s$ and at least $6wk$, and $k$ objects per image, Algorithm 3 with $L = 8\,w\,k$ and $\tau = \frac{k}{O(m\,2^k)}$ recovers all $m$ objects with $S = \Omega\left(2^k\,m\ln(m\,s)\right)$ samples with probability 9/10.*

**Proof**

As before, in order to prove this lemma, we must compute (1) $p_{\text{mid}}$, (2) the number of strings arising from a single object that we need to see, and (3) the number of problematic overlap strings we need to protect against. We have $p_{\text{mid}} = k/(16\,m\,2^k)$ from Claim 35.

From a single object, as before, there are: $4s/L$ required contiguous $L$-pixel portions. Over the $m$ total objects, then, there are $4ms/L$ strings of this type.

Similarly, the number of $k$-object $L$-pixel strings that could be problematic is the total number of tuples of $k$ objects, $k$ indices, and the front-to-back ordering.

**Claim 36** *The total number of tuples generating problematic overlap strings with $k$ objects is at most $\binom{m}{k} \cdot s^k \cdot k! \leq m^k s^k$.*

Applying the result of Lemma 29 and incorporating Claims 30, 36, 35, we have that the number of required samples is:

$$O\left(\max\left\{\frac{\ln(n_{\text{single}})}{p_{\text{mid}}}, \frac{\ln(n_{\text{po}})}{p_{\text{mid}}}\right\}\right) = O\left(\max\left\{\frac{\ln(4ms/L)}{p_{\text{mid}}}, \frac{\ln(m^k\,s^k)}{p_{\text{mid}}}\right\}\right) = O(2^k\,m\,\ln(m\,s)).$$

Finally, as before, due to Lemma 32, the ends of the objects are correctly recovered.

∎

## Appendix D. Proofs for Inference Algorithms

### D.1. Arbitrary Known Objects – DP Algorithm (Algorithm 5)

In this section we provide details of the dynamic programming algorithm that recovers a minimal explanation for an image arising from a known set of objects.

**Algorithm Summary**   We scan the image left to right. At each location of the image, for each potential source, where the "source" is either the background at appropriate location or the identity of the object and the location of the pixel in that object, we record the fewest objects required so far if we were to use that potential source. At any step, there are at most $s\,m$ options for the source, so the total size of the dynamic programming table is $s\,m \cdot d$.

When scanning, we either continue in the object we already were on, switch to a new object, switch to the background, or continue in the background. Upon switching, we either:

1. **Start New Object** We start at the beginning of a new object, cutting off the end of the object we were just in. To be in this case, we must be at location 1 of the proposed new object.
2. **End Current Object; Resume Object Underneath** We start in the middle of new object (behind), having finished the current one (on top). To be in this case, we must have just finished pixel $s$ of the object. We can start from any location of the new object.

Using this fact, the dynamic programming algorithm is going to behave as follows: assuming we've already filled out column $i$, we will fill column $i + 1$ out by first filtering out all the object locations that could match the $i+1$ bit in the image. Next, we check what happens if we try to start a new object. Since we are starting anew, we do not worry about the location in the previous instance. We pick the smallest number of objects needed so far out of any object/index combination at the previous level, compare it to the minimum number of objects needed if we are using background, choose the minimum of the two, and add 1 (since we are starting a new object). Otherwise, we take the minimum of continuing the existing object instance or a object behind emerging after the current object instance ends. Finally, we also update the background table by tracking how many objects would be required to use this bit of the background, if it matches.

**Theorem 37** *In either the open or closed room model, suppose an image is generated from our usual generative process except that objects all have the same length and are arbitrary. Then, given the description of the image, objects, and background, Algorithm 5 finds an explanation for the image, i.e., ascribes each pixel of the image to a pixel in an object, in a way that is (a) consistent with the generative model; (b) uses the minimum number of objects required to describe this image under our generative process. The size of the DP table is $\Theta(s\,m\,d)$ and the run time is $O(d\,s^2\,m^2)$.*

**Proof**

First, in the base case, where we are at pixel 1 of the image, we trivially find an explanation that only consists of 1 object instance. Thus, we solve the base case. Now, assume that at pixel $i$ in the image, for every object $o$ and every position $j$ in that object, we have stored the fewest objects needed to get there (with $\infty$ meaning impossible). We are interested in whether the proposed explanations we devise for the image up to the $i + 1^{\text{st}}$ pixel give the fewest possible object instances needed to produce the image so far conditioned on using a given source (a given location in a given object, or the appropriate location in the background) to explain pixel $i + 1$ in the image. By the inductive hypothesis, our table includes this information for pixel $i$. Any explanation for the first $i + 1$ pixels of the image must either have an object (or the background) continue from pixel $i$ to pixel $i + 1$, have an object end at pixel $i$, or have a new object start at pixel $i + 1$. Each of these cases is handled in lines 8, 9, and 11 in the DP algorithm and therefore the induction is maintained. The final explanation step takes the solution of overall least cost at the last pixel (conditioned on no object extending past the room boundary in the closed-room model). Thus, the solution our algorithm finds must be one with a minimal number of objects.

---

**Algorithm 5** Inference – Arbitrary Objects, DP Algorithm

---

1: **Input:** image of size $1 \times d$, description of objects and background
2: **Result:** explanation: source for each pixel of the image
3: $T \leftarrow$ table of size $(m + 1) \cdot s \cdot d$ elements, set to $\infty$
  $\{T[o, j, i]$ represents the fewest number of objects needed to produce the image up to index $i$ subject to pixel $i$ in the image being produced by pixel $j$ of object $o$; $\infty$ if not possible $\}$
4: $T[o, j, 0] \leftarrow 1$ if the $o^{\text{th}}$ object's $j^{\text{th}}$ position is the same as position 0 of the image. Do this $\forall j$ if open room model, else just for $j = 1$
5: $B \leftarrow$ table of size $d$ elements (only one dimension), set to $\infty$
  $\{B[i]$ represents the fewest number of objects needed to produce the image up to index $i$ subject to pixel $i$ being produced by pixel $i$ of the background; $\infty$ if it is not possible$\}$
6: $B[0] \leftarrow 1$ if the first pixel of the background matches the first pixel of the image
7:
8: **for** $i = 1, 2, ..., d$ (pixel in image) **do**
9:   **for** each $o, j$ such that the $o^{\text{th}}$ object's $j^{\text{th}}$ index is the same as $img[i]$ **do**
10:     **if** j == 1 **then**
11:        $T[o, j, i] \leftarrow \min\{\min_{o', j'}\{T[o', j', i - 1]\}, B[i - 1]\} + 1$
12:     **else**
13:        $T[o, j, i] \leftarrow \min\left\{T[o, j - 1, i - 1], \min_{o'}\{T[o', s, i - 1]\} + 1\right\}$
14:     **end if**
15:   **end for**
16:   **if** $B[i]$ matches current pixel of image **then**
17:     $B[i] \leftarrow \min\{B[i - 1], \min_{o}\{T[o, s, i - 1]\}\}$
18:   **end if**
19: **end for**
20: explanation $\leftarrow$ Backtrack starting at the minimum value $v$ in the $d^{\text{th}}$ matrix, $T[:, :, d]$. Store $o$ and $j$ having minimum $v$ for each pixel of image – if closed room, the second index of the start of the backtracking must be $s$, representing the end of the last image or it must come from background, and for the beginning of the image, we must be at pixel 0 of some object. if open room, there are no constraints
21: **return** explanation

---

The runtime arises from $d$ runs of outermost loop, then at most $m\,s$ runs of inner loop, and at most $ms$ checks within the loops.

$\blacksquare$

### D.2. Well-Structured Known Objects (Theorem 18)

**Theorem 18** *Whether the image has a distinct background (Defn. 4) or a well-structured background Defn. 5), the algorithm given in Algorithm 4 recovers an explanation for the image that is correct and explains at least $d - \Theta(wk^2)$ pixels and runs in polynomial time.*

**Proof** In order to prove this, we argue that for any indices $i_{\text{start}}$ to $i_{\text{end}}$, that match a single object, $i_{\text{start}} + k\,w$ to $i_{\text{end}} - k\,w$ must indeed arise from the object.

**Definition 38** *A* pure segment *is a block of contiguous pixels that in the ground truth image, comes from a single object or the background.*

**Claim 39** *There are at most $2k + 1$ pure segments in an image.*

**Proof** Each boundary between two pure segments is the beginning or end of an object. There are at most $2k$ beginnings and ends, so there are at most $2k + 1$ segments. $\blacksquare$

### Lemma 40

*Assuming objects are $w$-well-structured, then if the algorithm sees a segment of size $\ell \geq 2wk+1$ that matches a single object $o$, then the middle $\ell - 2(w-1)(k-1)$ pixels must be a pure segment arising from object $o$.*

**Proof** In this proof, we use the word "object" liberally, including in situations where the background is well-structured and has a large visible chunk. Suppose we see an $\ell \geq w$ pixel segment that matches a single object $o$. Since we wish to be correct on any pixel that we ascribe to a particular object, we need to account for the possibility that this $\ell$-pixel segment contains overlaps.

Let us start by showing the following claim:

**Claim 41** *Suppose we have a segment $s$ of size $\ell \geq 2wk+1$ that matches $\ell$ pixels of a single object $o$. If in $s$, pixel $i$ and pixel $j$ indeed arise from the same instance of object $o$, then for all $q \in [i, j]$, pixel $q$ of the image must also arise from the same instance of object $o$.*

**Proof** We can use a stack to model the creation of the image. At the index at which an object begins, we push the object onto the stack, and when it ends, we pop it from the stack. This corresponds exactly to seeing a new object placed on top of the existing image. Then, when reading an image left-to-right, the object on the top of the stack at that point is the object from which the current pixel comes from. Thus, if two distinct pixels, pixel $i$ and pixel $j$ both come from the same object, then there are two options for the configuration of the stack during that time:

1. $i$ and $j$ come from the same copy of object $o$ and object $o$ has been on the top of the stack throughout.

2. $i$ and $j$ come from the same copy of object $o$, object $o$ was on top, then other objects were added *and* removed before index $j$.

We show that (2) is not possible. In particular, suppose that for contradiction (2) occurred; let $o' \neq o$ be the object that was added and removed with no other object added on top. Such an $o'$ must exist by the stack property. By the first property in the definition of well-structuredness, if an object has come and gone, at least $w$ pixels have transpired, so at least $w$ pixels of that object coincide with $s$, which matches $\ell$ pixels of a different object. This violates well-structuredness, so (2) is not an option. Thus, it is clear that the claim holds. ∎

Now, all that remains to be shown is that there exist these indices $i$ and $j$. To show this, we consider the possibility of having overlaps constituting the two ends of the string of length $\ell$ and show that once $\ell$ is big enough, these indices $i, j$ must exist.

By Pigeonhole principle, in the left half of the segment, there must be at least one object that has at least $w$ pixels visible. Likewise, in the right half of the segment, there must be at least one object that has at least $w$ pixels visible. Now since $s$ exactly matches $\ell$ pixels of object $o$, and due to well-structuredness, both these segments of at least $w$ pixels must come from the same instance of object $o$ (because there is exactly 1 location in object $o$ that would have these pixels). This implies there are at least two indices $i, j$ that arise from the same instance of $o$, and so the claim applies. By well-structuredness, if the leftmost end starts with a different object $o_1$, then at most $w - 1$ pixels at the leftmost end that can be from $o_1$ before it must switch to some other object $o_2$. This can go on at most $k - 1$ times if the background is distinct and $k$ times if the background is well-structured, so at most $(k)(w - 1)$ pixels from the left end can be from other objects masquerading. The same holds from the right. Therefore a segment of at least $\ell - 2(w - 1)(k)$ will be ascribed to $o$ by the algorithm.

∎

Given this lemma, we know that for each pure segment we fail to assign at most $2wk$ visible pixels. Thus, the total number of pixels for which the explanation is unassigned is at most $2wk(2k + 1) = \Theta(k^2 w)$.

**Runtime** Naively, this takes at most $O(m \, d \, s^3)$ time: we consider each string of length $z$ in the image and see if it matches any of segments of length $z$ in the $m$ images. Thus, for each $z = w...s$, it takes $(d - z + 1) \cdot (s - z + 1) \cdot m \cdot z$, assuming comparing whether two strings of length $z$ takes time $z$. If we run this procedure for each of the $O(s)$ values of $z$, then the total cost is at most: $O(d \, s^3 \, m)$. This can be improved by using better string matching. For example, using FM index as described in Theorem 3 of [Ferragina and Manzini (2000)](#), with $O(ms)$ preprocessing time, we can look up a string of length $z, z \in [w, s]$ in time $z + 1 \cdot \log ms$. There are $d - z + 1$ strings of length $z$ in the image. Summing, we get:

$$\sum_{z=w}^{s} (d - z + 1)(z + \log^2 ms) \leq \sum_{z=w}^{s} dz + d \log^2 ms \leq O(ds^2 + ds \log^2 ms).$$

Thus, including preprocessing, the total time is $O(ms + ds^2 + ds \log^2 ms)$.

We can also solve this using $O(k)$ instances of finding the longest common substring between the remaining image and the set of objects. The fastest algorithm for longest common substring

involves using suffix trees and would require time $O(d + ms)$, making the overall runtime $O(kd + m s k)$.

■

### D.3. Noisy Images – Theorem 21

#### D.3.1. EXAMPLES THAT BREAK EXACT-MATCH ALGORITHMS

We present two examples, one that breaks the dynamic programming algorithm, and the other that breaks any exact-match-based algorithm.

**Example That Breaks Exact DP**    Suppose the three colors used are $A, B, C$. Let the background be $CBABABABABA...$ with objects $BAAAA$ and $ABBB$. If we place the first object on the background one pixel in, we will get the following image: $CBAAAABABABA...$. Now suppose an adversary corrupts the second pixel of the image from a $B$ to an $A$ so the image now looks like $CAAAAABABABA...$. Then, the exact DP algorithm would explain the image using a copy of object 2 for each pixel that is an $A$ and then placing one of the objects every two pixels to get the $AB$ sections. Thus, just corrupting one pixel will cause the dynamic programming algorithm to go from giving an optimal solution to one that requires something like $d/2 + 2$ objects to explain the $d$ pixels.

**A Family of Examples with Well-Structured Objects**    Suppose we are in the open room model. We have two colors and well-structured objects $A$ and $B$, where object $A$ starts with the first color and object $B$ with the second color. Both objects are of length $d$. In this setup, we can create any image of length $d$ by simply placing object $A$ on top when first color is present and object $B$ on top when the second color is present. In order to generate the image, pick one of the objects and place it on the canvas so it is fully visible. Next, pick a location $i$ in the middle $d/2$ pixels of the image and flip the bit in that location. This completes the adversarially-corrupted image. We can show that we will require at least $d/(4w)$ objects to exactly explain this image. First, we know that the first $d/4$ bits exactly match the chosen object, as do the last $d/4$ bits, and due to well-structuredness they must each either arise from the exact same set of indices as they did originally or from problematic overlaps. Suppose they did not arise from problematic overlaps but rather from the same set of indices. Then, in order to get something in the middle that does *not* match that object, a different object must have started and ended. However, since the only available objects are of size $d$, this is not possible. Thus, at least one side must have been composed of problematic overlaps. In particular, since no segment of length $w$ can have arisen from the correct object unless the indices match, it must be the case that there is a transition between objects at least once every $w$ pixels. Thus, there are at least $d/(4w)$ objects required to explain that portion of the image, for a total of at least $d/(4w)+1$ objects to explain an image that would require just 1 object if corruptions could be accounted for.

#### D.3.2. THEOREM PROOF

**Theorem 21** *Suppose an image is generated from $\epsilon$-strongly, $w$-well-structured objects, and an adversary of strength $(\alpha, W)$, $\alpha < \epsilon/4$, acts on it. Then, Algorithm 4 with the modifications in*

*parentheses would recover in polynomial time an explanation for the unadulterated image that is correct in $d - \Theta(w_{alg}k^2)$ pixels.*

**Proof**

As before, our argument hinges on the algorithm's ability to find and correctly identify signatures. In this case, finding a signature requires a bit more work than looking for an exact match. To show that our algorithm succeeds in identifying the source of a large number of pixels of the image, we crucially argue that if a large chunk in the image seems most identical to a chunk from a single object, a subset of it originally arose from that object (prior to corruptions). To get there, we first show that an object that is $\epsilon$-strongly, $w$-well-structured is also $\epsilon/2$-strongly, $w_{\text{alg}}$ well-structured. Next, we show that for a range of $\alpha$ values (representing the strength of the adversary), the correct source object is identifiable even after corruptions. Finally, we argue the key lemma described above.

**Lemma 42** *If a set of objects is $\epsilon$-strongly, $w$-well-structured, then for any $z > w$, the set of objects is $\epsilon/2$-strongly, $z$-well-structured.*

**Proof**

From the definition of $\epsilon$-strongly $w$-well-structured, we know that there do not exist two strings $s_1, s_2$ of length $w$ such that $1 - \epsilon$ fraction of the pixels match. This implies that for all strings $s_1, s_2$, of length $w$ at most $1 - \epsilon$ fraction of the pixels match, and so for all strings $s_1, s_2$, of length $w$ at least $\epsilon$ fraction of the pixels differ. Now we consider strings $s'_1, s'_2$ from such objects of length $z > w$. Write $z = s\,w + j$, where $j < w$ and $s \geq 1$. Then we know that at least $\epsilon sw$ pixels of $s'_1$ differ from $s'_2$. Then, we have that the fraction of pixels that differ is at least:

$$\frac{\epsilon sw}{s\,w + j} > \frac{\epsilon sw}{(s+1)w} \geq \frac{\epsilon}{2}.$$

And so, we have shown that for all $z > w$, a set of objects that is $\epsilon$-strongly, $w$-well-structured is $\epsilon/2$-strongly, $z$-well-structured.

∎

**Lemma 43**

*Suppose objects are $\epsilon$-strongly, $w$-well structured, and an adversary with strength $(\alpha, W)$ where $\alpha < \epsilon/4$, acts on the image. Then if a segment in the image of length $\ell \geq \max\{w, W\}$ arises from positions $i$ through $i + \ell - 1$ of a single object, then it is closer to the substring from $i$ to $i + \ell - 1$ of that object than it is to any other segment of length $\ell$ of that object or any other object. That is, if a segment is $\alpha$-approximately close to a segment of object $o$, it cannot be $\alpha$-approximately close to any segment from any other object $o'$ or any other segment from object $o$.*

**Proof**

There are two cases for us to consider. In the first case, $W > w$. In this case, from the previous lemma, we know that the objects are also $\epsilon/2$-strongly, $\ell$-well-structured.

Let the string in the image be $s_1$ which is generated by corrupting $\alpha$ fraction of the pixels of a string of length $\ell$ starting from index $j$ of object $A$, which we call $s_\star$. Then, the Hamming distance between $s_1$ and $s_\star$ is at most $\alpha \ell < \frac{\epsilon}{4} \ell$. The Hamming distance between any other substring of an

object of length $\ell$ denoted $s_{\text{any}}$ and $s_\star$ is at least $\frac{\epsilon}{2}\ell$. Thus, by the triangle inequality $d(s_1, s_{\text{any}}) \geq d(s_\star, s_{\text{any}}) - d(s_1, s_\star) > \frac{\epsilon}{2}\ell - \frac{\epsilon}{4}\ell = \frac{\epsilon}{4}\ell$.

In the other case, $w > W$. Then, at most $\alpha W < \alpha w < \frac{\epsilon}{4}w < \frac{\epsilon}{4}\ell$ pixels are corrupted and the same argument using triangle inequality holds: any other string of length $\ell$ would be at least $\epsilon \ell$ pixels different from this one.

∎

With this lemma, we have that any $w_{\text{alg}}$-pixel (with an appropriate fraction possibly corrupted) string from an object will be closer to its source object than anything else. This allows us, in analysis, to be able to uniquely identify the source of a large piece. This will be essential in arguing that if enough pixels of an object are visible in the image, then the algorithm will recover part of that object.

**Lemma 44** *Assume objects are $w$-well-structured and the adversary has strength $(\alpha, W)$, where $\alpha < \epsilon/4$. Let $w_{alg} = \max\{w, W\}$. Then if the algorithm sees a segment of size $\ell \geq 2w_{alg}k + 1$ that approximately matches (as in Defn. 20) $\ell$ pixels of a single object $o$, then the middle $\ell - 2(w_{alg} - 1)(k - 1)$ pixels must be a pure segment arising from object $o$.*

**Proof** Again, as before, we first analyze the possibility that problematic overlaps comprised this image.

**Claim 45** *Suppose we have a segment $\tilde{s}$ in $\tilde{\mathcal{I}}$ of size $\ell \geq 2w_{alg}k + 1$ that $\alpha$-approximately matches that many pixels of a single object $o$. If in $s$, the corresponding segment pre-corruption, pixel $i$ and pixel $j$ indeed arise from the same instance of object $o$, then for all $q \in [i, j]$, pixel $q$ in $s$ of the image $I$ must also arise from the same instance of object $o$.*

**Proof** This follows from the same argument as before, except now instead of applying well-structuredness to argue that an object cannot have been added and removed to the stack described in the proof of Claim 41, we apply $\epsilon$-well-structuredness to argue it. Once again, we must consider the same two potential explanations for the matching between object $o$ and the corresponding locations in string $s$ described previously. Either $i, j$ come from the same copy of object $o$ and object $o$ has been on the top of the stack throughout, or $i, j$ come from the same copy of object $o$, object $o$ was on top, then another object $o'$ was added and removed before index $j$. The latter would necessitate that at least $w$ pixels have come and gone. In the context of the original problem, in order to be in this case, we would need that the string $\tilde{s}$ $\alpha$-approximately matches $o'$. However, by Lemma 43, we know that any object that isn't $o$ must be further away than $\alpha \cdot w$ from the object in at least one window. This concludes the proof of Claim 45.

∎

Based on this, we may continue proving Lemma 44. By the Pigeonhole principle, in the left half of the segment, there must be at least one object that has at least $w_{\text{alg}}$ pixels visible. Likewise, in the right half of the segment, there must be at least one object that has at least $w_{\text{alg}}$ pixels visible. Now, since $\tilde{s}$ approximately matches $\ell$ pixels of object $o$ and due to $\epsilon/2$-strongly well-structuredness, both these segments of at least $w_{\text{alg}}$ pixels must come from the same instance of object $o$. This implies that there are at least two indices $i, j$ in $s$ that arise from the same instance of $o$. From Claim 45, we know that if we have these pixels $i, j$ in $s$, then all of the pixels of $s$ in the middle arise from the

same instance of that object, and so all of the pixels of $\tilde{s}$ must also arise from object $o$, albeit with a limited number of corruptions.

Finally, we must analyze the length of the longest possible segment between $i, j$. We could lose $w_{\mathrm{alg}} - 1$ pixels $k$ times on each end, which leads to the bound in the statement of Lemma 44. ∎

From this lemma, we have that if a large enough segment of an object is seen, then all but $\Theta(w_{\mathrm{alg}}k)$ pixels of it are correctly identified with the source object. Since there are at most $k$ objects, the algorithm recovers a correct explanation for at least $d - \Theta(k^2 w_{\mathrm{alg}})$ pixels.

**Runtime** Naïvely, to look up whether a string of length $z$ $\alpha$-approximately matches any string in the objects, we must look through up to $(s - z + 1) \cdot m$ strings and compare pixel-by-pixel, requiring $(d - z + 1) \cdot z \cdot (s - z + 1) \cdot m$. Thus, summing over values for $z$, this takes at most $O(d\,s^3\,m)$.

In order to improve this algorithm, we can improve the way we find the largest pure segment still unexplained as follows. Consider the following dynamic programming table: it contains $d \times s$ cells, and each cell contains $w_{\mathrm{alg}}$ entries. The entry $i$ in cell $j, h$ represents the number of corruptions required in the last $i$ pixels to make pixel $h$ of the object the explanation for pixel $j$ in the image. To update entries in a cell in this table, we check the cell for the previous pixel in the same object and the previous pixel in the image (i.e., cell $j - 1, h - 1$). Based on where the corruptions are in this diagonally previous cell, we update the entry $i$ in the current cell. In particular, if pixel $h$ is not a correct explanation for pixel $j$, then if there were $x$ mistakes in the $i - 1$ location in cell $j - 1, h - 1$, there are $x + 1$ mistakes in the $i$ location of cell $j, h$. If a stored value grows beyond $\alpha\,w_{\mathrm{alg}}$, then we may terminate that chain. In terms of initialization, for each cell in the first row and for each cell in the first column, we fill in the entry asking how many pixels are incorrect in the last 1 pixel with either a 0 or a 1, depending on whether that location can be explained by that pixel in the object or not. From here, we can continue the procedure as described above. We choose the longest diagonal chain that is never broken as the largest possible single object and continue with the algorithm.

This DP table has size $O(dsw_{\mathrm{alg}} \log w_{\mathrm{alg}})$ and each entry can be filled in constant time. We need $m$ copies of it per pure segment detection and then need to use to use up to $2k + 1$ instances of it, since that is an upper bound on the number of pure segments we have. Thus, the runtime is $O(ds\,m\,k\,w_{\mathrm{alg}})$.

This concludes the proof of Theorem 21. ∎

## Appendix E.  Computational Hardness of the Learning Task

In this section, we define a related decision version of our problem and show that it is NP-hard via reduction from set-splitting (also known as hypergraph 2-coloring) where the sets (hyperedges) are of size 2 or 3. We start by formally defining the two problems. Then, we describe our reduction.

**Object Learning Problem Definition** Given an alphabet $\Sigma$, a background color $b \in \Sigma$, an integer $n$, and a set $S$ of $m$ triples $(\mathcal{I}_i, l_{i1}, l_{i2})$, where $\mathcal{I}_i \in \Sigma^d$ and $l_{i1}, l_{i2} \in \mathbb{Z}$, does there exist a set $T$ of $n$ objects, i.e., strings $o_j \in (\Sigma \setminus \{b\})^*$, such that for each string (image) $\mathcal{I}_i$ there exist two objects $o_{j1}, o_{j2} \in T$ of length $l_{i1}, l_{i2}$ respectively such that $\mathcal{I}_i$ can be produced using $o_{j1}, o_{j2}$ in the open-room model? (By "produced in the open room model" we mean through the process defined in Section 2.3).

**Note:** The above is essentially the decision-version of the problem of learning objects from images with (exactly) two objects per image, with the added information of the lengths of the objects in each image. Note that the algorithm of Section 3.2 for learning objects in the partially-random model with well-structured objects is only helped if it is given the lengths of the objects since it finds the objects exactly and can always ignore this information.

**Set-Splitting Problem Definition**    Given a set of $n$ items (variables) and a collection of $m$ subsets of these variables (clauses), where each clause is of size two or three, can we assign variables to 0 (false) or 1 (true) such that no clause has all its variables 0 or all 1? This problem is known to be NP-hard (Garey and Johnson (1990), Appendix 3.1, [SP4], Comment).

**Reduction Construction**    First, before getting to the instance of set-splitting, it will be helpful to define the underlying $n \times O(n^3)$ variable-clause incidence matrix $M$ with one row for each of the $n$ variables, one column for each of the $\binom{n}{2} + \binom{n}{3}$ possible clauses of size 2 or 3, and setting $M_{ij} = 1$ if variable $i$ belongs to clause $j$ and $M_{ij} = 0$ if it does not. Note that at this point we are considering all possible clauses, not just the clauses in the given set-splitting instance.

The underlying indicator vector $\mathbb{I}(x)$ corresponding to variable $x$ is simply the row of matrix $M$ corresponding to $x$ prepended with a 0 at the start to make some bookkeeping easier later.

From this, we generate $\mathbb{I}_r(x) = \mathbb{I}(x)$, and $\mathbb{I}_l(x)$ which is the mirror of $\mathbb{I}(x)$; these are the right and left indicator vectors respectively. With these indicator vectors, we can now generate the images we need for casting the set-splitting problem as an object learning one.

To convert an instance of set-splitting, which we shall denote $\mathcal{SS}$, to an instance of object learning, we now generate three types of images over an alphabet of 6 different colors $\{$ 0, 1, $T$, $F$, $b$ (background), $g$ (grey)$\}$.

**Variable Images.**    First are what we call the "variable images." For each variable $x$, we generate two images:

- background ($b$), followed by indicator vector $\mathbb{I}_l(x)$, followed by $T$, followed by indicator vector $\mathbb{I}_r(x)$, followed by background ($b$).

- background ($b$), followed by indicator vector $\mathbb{I}_l(x)$, followed by $F$, followed by an indicator vector $\mathbb{I}_r(x)$, followed by background ($b$).

One pixel of background suffices on the left and right above. In future, wherever there are background pixels, we include enough of them to ensure the image has size $2 \cdot \binom{n}{2} \cdot \binom{n}{3} + 3$.

The sizes of each of the two objects in these images is $\binom{n}{2} + \binom{n}{3} + 2 = \frac{n^3-n}{6} + 2$.

**Set Images.**    The second type of image is the "set image." For each set in the problem instance, suppose its index is $k$ (i.e., it corresponds to column $k$ in matrix $M$). Then, we generate two images:

- 1, followed by $k$ grey ($g$) pixels, followed by $T$ followed by background.

- 1, followed by $k$ grey ($g$) pixels, followed by $F$, followed by background.

The objects for these images are of size $k$ and $\binom{n}{2} + \binom{n}{3} + 2 = \frac{n^3-n}{6} + 2$.

**Mask Images.** The final kind of image is what we call "mask images." For each value $j$ from 2 to $\frac{n^3-n}{6}$, these images are background, followed by $j$ pixels of grey, followed by background, followed by 1 pixel of grey. The sizes of the objects are $j$ and 1.

This completes the reduction. Let us call this instance of object learning, i.e., determining whether there exists a set of $2n + \binom{n}{2} + \binom{n}{3}$ objects that can be used to generate the aforementioned set of images, $\mathcal{OL}$. We now claim this reduction is correct, that is, a solution to $\mathcal{OL}$ exists if and only if a solution to $\mathcal{SS}$ exists.

**Lemma 46** *A solution to $\mathcal{OL}$ exists if and only if a solution to $\mathcal{SS}$ exists.*

**Proof** We first describe how the existence of a solution to $\mathcal{SS}$ implies a solution to $\mathcal{OL}$ ($\Leftarrow$) by constructing a sufficient set of objects that produce the desired images.

Given a solution to $\mathcal{SS}$, we create a set of objects that solve $\mathcal{OL}$ as follows. If a variable $x_i$ is true in the $\mathcal{SS}$ solution then the two associated objects are (1) the indicator vector $\mathbb{I}_l(x_i)$ followed by $T$, and (2) $F$ followed by the indicator vector $\mathbb{I}_r(x_i)$. On the other hand, if the variable is false, then the two objects are (1) indicator vector $\mathbb{I}_l(x_i)$ followed by $F$, and (2) $T$ followed by the indicator vector $\mathbb{I}_r(x_i)$. Thus, whether $T$ or $F$ is associated with the left indicator vector depends on whether the variable is true or false, respectively. Note that the object lengths also work out correctly. Next, from the mask images, we create grey masks of size $k$ and size 1 for the respective images. Finally, no additional objects are required for the set objects: since the variable settings constitute a solution to the set-splitting instance, we know that there must be at least one variable in the set that is true and at least one that is false. As a result, for a given set, we create the two images associated with it as follows: we can take object (1) for the true variable and mask of appropriate size to create the first set image, and we take object (1) for the false variable and mask of appropriate size to create the second set image. Therefore, we have a solution using the correct number of objects, $2n$ variable objects and $\binom{n}{2} + \binom{n}{3}$ mask objects and no additional objects for the set images.

For the reverse direction ($\Rightarrow$), we show that a solution to $\mathcal{OL}$ implies a solution to $\mathcal{SS}$.

Consider a solution to $\mathcal{OL}$. Observe that for variable images, the stipulation about object sizes mandates that the left indicator vector for that variable followed by a single character is one object, and the other object must be a single character followed by the right indicator vector for that variable. Since the indicator vectors are different for each variable, this requires a total of at least $2n$ distinct objects, and it can only be done with $2n$ objects if exactly two objects are used per variable. In particular, requiring exactly two objects per image implies that if for some variable $x$, $\mathbb{I}(x)$ followed by $T$ is one object, we cannot have $\mathbb{I}(x)$ followed by $F$ as another object and vice-versa, implying that we will not generate objects that indicate both true and false for a particular variable. Likewise, for each mask image, the two objects are the $k$ grey pixel block and the single grey pixel, for a total of $\binom{n}{2} + \binom{n}{3}$ mask objects. Note that any solution to $\mathcal{OL}$ must not use any additional objects for the set images.

Now, consider the two set images corresponding to set $k$. We can generate these using existing variable and mask objects iff we have a variable object that has a 1 at distance $k + 1$ from a $T$, and a variable object that has a 1 at distance $k + 1$ from an $F$. Since the only such variable objects are those corresponding to variables in set $k$, this means that if we set variable $x$ to true or false based (respectively) on whether its indicator vector followed by T or F is one of the variable objects, we will have a legal solution to the given set-splitting instance $\mathcal{SS}$ because every set will have at least one variable in it set to true and at least one variable set to false (and no variable will be set to both true and false since there are only $2n$ variable objects). ∎

