# OpenReview forum: "A Model for Combinatorial Dictionary Learning and Inference"
_algorithmiclearningtheory.org/ALT/2025/Conference — ALT 2025_

### Official Review · Reviewer_YwbN · 2024-11-08

**Rating:** 7
**Confidence:** 4

**Review:**

This paper presents algorithms for learning and inference under a non-linear version of dictionary learning problems. The authors consider a one-dimensional model where localized atoms (of small support) can be placed on a larger canvas in different locations while occluding one another. This resembles the way objects are composed on an image, whereby foreground objects occlude those in the background. The authors consider different version of this model (with different degrees of randomization) and study greedy-like algorithms to learn a set of prototype objects (or atoms) as well as inferring their locations (or segmentation) in a given sample. The authors put forward a key assumption that they term "well-structuredness", which while weak allows them to provide correctness and convergence of the studied algorithms.

### Pros:

* This paper studies a non-linear version of sparse coding problems which, to my knowledge, has not been studied so far.
* The analysis is rigorous, and the obtained results are nice and useful - i.e. certifying the correctness of the proposed algorithms in polynomial time under mild conditions.
* The paper is mostly well written and nice to read.

### Cons

* I think the text can be improved slightly without much effort by revising some parts. (see below)
* While not necessary and orthogonal to the presented results, it would have been nice to have at least some numerical demonstration on some of the problems referred to in the motivation.

### Questions and suggestions

* I find the definition of $w$-well-structured very interesting, which the authors relate to analogous quantities of coherence (popular in the sparse coding literature). This notion, however, is defined on a finite discrete space, and requires "exact" matching. Even their $\epsilon$-strong version requires that no objects have substrings that match (exactly) in $1-\epsilon$ fraction of their pixels. Generalizing these notions to sequences over real values, this implies and exact match (with zero error) between the subsequences (in a $1-\epsilon$ fraction of the pixels). I wonder if the authors could consider, or at least comment on, extensions to these continuous settings where one could allow a bounded error among substrings, given by a bounded $\ell_p$ norm. I'm tempted to infer that bounded $\ell_0$ "norm" is already the case considered here.

* On page 5, second paragraph: "The object with the highest depth is placed in row k, ..". I don't think this object needs to be unique. Does this matter?

* The section 2.5 on sequencing methods is a nice introduction to the algorithms they propose later. However, they directly start talking about "segments" without defining them formally (unlike the previous formal definitions of objects, background, scene, etc). Thus, it's a bit unclear what the authors mean. This could be easily fixed by simply making this more precise.

* I'm a bit confused by Lemma 11, and the subsequent results. There, I see that the probability of an event of relevance (seeing a L-long segment of an object... ) is at least $(1- c_1)^{k-1} * c_2 * k/m$. However, if $d\approx d'$, and $k \ll m$, i don't see how this probability could be arbitrarily controlled. What am I missing?
Related to this, in Theorems 12 and 14, could the authors present their results with a parametric form of the failure probability?

* On Section 4, the authors explain in words what the sparse coding problems is in this setting. Though, since this is the first work considering this problem formulation, it might be nice to actually flash it out analytically (e.g. minimizing the number of chosen atoms so that reconstruction is achieved)

*

Minor:

* The text might benefit from some slight revision; e.g. avoiding single-sentence paragraphs (like the two ones on the first page).
* $k<<$ -> $k\llm$ on page 2.
* First sentence in Assumptions section (page 3) might have a typo (or an extra "there"?)
* In Section 2.3, point 5, i don't think the scene operator S has been defined yet.
* Text in Fig 1 is too small to be readable.
* Why are $\epsilon$ and $\alpha$ bolded on Page 12?
* Because the atoms that the authors consider have small support and can be placed on different places of the canvas, to me this looks more like a non-linear version of a *convolutional* dictionary learning problem - see e.g. [Papyan et al, "Working locally thinking globally: Theoretical guarantees for convolutional sparse coding"]. The authors might want to comment on these similarities.

**Paper Award:**

No

---

> ### Author Response · Authors · 2024-11-25
>
> Re $\ell_p$ norms – That’s an interesting question, and we have only ever though briefly about it. Here’s the difficulty that arises that we’d need to work through:
> Our assumption is that objects are fixed strings. In the new setting, if it’s the case that when you see the same object twice, both instances look the same, our results should still hold; the main addressed issue in our current analysis is ensuring strings from different objects don’t look the same (what we called problematic overlaps). On the other hand, if each time a string appears it looks mostly the same but with slight variations, then we have to worry about mistakes of the opposite kind, i.e., the same string looking different in different appearances. This could be a great direction for future work!
>
> Uniqueness – The objects do have to be distinct – this is because if they are not, then the probability of seeing a problematic overlap string can go up substantially based on the “multiplicity” of the objects.
>
> Lemma 11 – We only need this to be bounded away from 0, since this controls sample complexity. Defining this probability as a, you can see that in Theorem 12, the sample complexity depends on 1/a. As long as k is relatively small, this is indeed bounded away from 0. The results translate to with probability $1-\delta$ with the standard $\log(1/\delta)$ multiplicative increase in sample complexity.
>
> We will define “segments” and the problem statement in Section 4 formally. Thanks for the suggestions! Thanks also for the comments and pointer to paper in the “Minor” section; we will address these.

---

### Official Review · Reviewer_ado3 · 2024-11-09
**Good attempt at modeling non-linear dictionary learning with severe limitations**

**Rating:** 5
**Confidence:** 3

**Review:**

The article presents a novel combinatorial model for non-linear dictionary learning and inference, focusing on an occlusion model for discrete objects in a 1D image. The authors introduce the concept of "well-structuredness" to ensure the uniqueness of components and demonstrate its sufficiency for learning and inference tasks. The work also explores adversarial robustness, providing insights into noise tolerance in combinatorial dictionary learning.

The article's strengths include its originality in proposing the combinatorial model and the well-structuredness property, its clear and concise presentation of the model and algorithms, and its significance in providing a theoretical framework for studying dictionary learning in non-linear settings. The authors provide comprehensive proofs for their claims and offer a reasonably detailed discussion of related work.

However, the work has some serious limitations. The authors acknowledge that their model is a simplification of real-world image generation and is currently restricted to one-dimensional images with quantized colors. The analysis also assumes specific conditions on object sizes and placement, which might not always hold in practice.
In addition, the authors do not discuss the tightness of the proposed bounds, particularly the exponential dependence on the number of objects in Theorem 15, nor do they provide lower bounds to support their findings. Including such discussion and providing lower bounds would significantly strengthen the results and offer a more complete understanding of the combinatorial dictionary learning problem.

**Paper Award:**

No

---

> ### Author Response · Authors · 2024-11-25
>
> Re simplified – Indeed, our work focuses on the one-dimension, quantized case. However, we believe this is already an interesting, non-trivial, and fundamental problem. It is natural to study how to recover dictionary elements when a compound scene has been formed from placing several small components in an overlapping manner.
>
> Re conditions in analysis --  Our goal is to provide an abstraction in which we can study combinatorial dictionary learning in this form and to give algorithms with provable guarantees under fairly reasonable assumptions.  For example, we think the partially-random model is quite reasonable (certain objects might tend to always be in the background so we do not require the front-to-back ordering be random, just the left-to-right placement).  Regarding the exponential dependence on the depth k, this is necessary because for an object that is always in the background, to see a given part of it one needs that none of the nearer objects are occluding it (see also discussion below). Certainly we make simplifications, but we view our work as a first step toward theoretical understanding of the problem of extracting objects from images. Given what AI can do, we ought to have a theoretical understanding.
>
> Re tightness and lower bounds – Please note the NP-hardness of the basic problem in the absence of assumptions; we prove this result in Appendix E. Also, we justify the reasons for exponential dependence at the top of page 8 (just before Section 3.2). In particular, since we allow for depth to be arbitrary, if there is an object that always appears at the very back, then we must account for the probability that all of the other objects will occlude some section of it.

---

### Official Review · Reviewer_cdrB · 2024-11-09

**Rating:** 6
**Confidence:** 2

**Review:**

Summary:

This paper studies the problem of learning objects from randomly generated images.




Strengths:

- This problem seems well-motivated.



Weaknesses:

- The paper focuses on one-dimensional, quantized data, which limits applicability to real-world, multi-dimensional scenarios.

- In terms of presentation, it would be helpful to define the terms more rigorously in mathematical terms.
For example, in the notation section, it would be clearer if it were self-contained.
If I am not mistaken, a canvas is a vector or sequence, yet I do not think it is formally defined.

**Paper Award:**

No

---

> ### Author Response · Authors · 2024-11-25
>
> Indeed, our work focuses on the one-dimension, quantized case. However, we believe this is already an interesting, non-trivial, and fundamental problem. It is natural to study how to recover dictionary elements when a compound scene has been formed from placing several small components in an overlapping manner.
>
> Thanks for pointing out that “canvas” is not formally defined – we will fix this and check for other similar things.

---

### Meta-Review · Area_Chair_oobc · 2024-12-07

**Recommendation:** Accept
**Confidence:** 4

**Metareview:**

The paper proposes a model for a non-linear version of dictionary learning. It is a one-dimensional model where objects of small support are placed on a large canvas (a vector). The objects may occlude each other, making the learning problem difficult. Given images generated from a common set of objects at random locations, the goal is either to reconstruct the objects or to explain how (a subset of) objects appear in a new image.

The key to all the proposed procedures is a well-structuredness assumption. This assumption says that each object has a sufficiently long and unique substring that serves as the signature of the object. The signature is used by the algorithms to identify each object in the images, and then the different parts of the object from different images can be put together to recover the object.

Pros:
- The model is novel and natural.
- Efficient algorithms are proposed.
- Theoretical guarantees are provided.
- The paper is mostly well-written.

Cons (and suggestions by the referees):
- The writing can be made more mathematical. I was confused with all the terms canvas, objects, depth, horizontal, etc. until page 5. It is nice to use the natural language to provide the intuition, but I think it would be better to define terms mathematically from the beginning of Section 2.
- The connection to the shotgun sequencing literature can be expanded and discussed earlier in the paper, as the problems are similar. The occlusion between objects appears to be a new feature of the combinatorial dictionary learning problem.
- The model is arguably too simple compared to the practical problem that motivates it. The well-structuredness assumption is quite strong. But I think the paper provides a good starting point of a potential thread of research.

**Paper Award:**

No